# Neural synchronization is strongest to the spectral flux of slow music and depends on familiarity and beat salience

Kristin Weineck[1,2]*, Olivia Xin Wen[1], Molly J Henry[1,3]

[1]Research Group "Neural and Environmental Rhythms", Max Planck Institute for Empirical Aesthetics, Frankfurt am Main, Germany; [2]Goethe University Frankfurt, Institute for Cell Biology and Neuroscience, Frankfurt am Main, Germany; [3]Department of Psychology, Toronto Metropolitan University, Toronto, Canada

**Abstract** Neural activity in the auditory system synchronizes to sound rhythms, and brain–environment synchronization is thought to be fundamental to successful auditory perception. Sound rhythms are often operationalized in terms of the sound's amplitude envelope. We hypothesized that – especially for music – the envelope might not best capture the complex spectro-temporal fluctuations that give rise to beat perception and synchronized neural activity. This study investigated (1) neural synchronization to different musical features, (2) tempo-dependence of neural synchronization, and (3) dependence of synchronization on familiarity, enjoyment, and ease of beat perception. In this electroencephalography study, 37 human participants listened to tempo-modulated music (1–4 Hz). Independent of whether the analysis approach was based on temporal response functions (TRFs) or reliable components analysis (RCA), the spectral flux of music – as opposed to the amplitude envelope – evoked strongest neural synchronization. Moreover, music with slower beat rates, high familiarity, and easy-to-perceive beats elicited the strongest neural response. Our results demonstrate the importance of spectro-temporal fluctuations in music for driving neural synchronization, and highlight its sensitivity to musical tempo, familiarity, and beat salience.

**\*For correspondence:**
kristin.weineck@ae.mpg.de

**Competing interest:** The authors declare that no competing interests exist.

## Editor's evaluation

This study investigated the neural tracking of music using novel methodology. The core finding was stronger neuronal entrainment to "spectral flux" rather than other more commonly tested features such as amplitude envelope. The study is methodologically sophisticated and provides novel insight on the neuronal mechanisms of music perception.

## Introduction

Neural activity synchronizes to different types of rhythmic sounds, such as speech and music (*Doelling and Poeppel, 2015*; *Nicolaou et al., 2017*; *Ding et al., 2017*; *Kösem et al., 2018*) over a wide range of rates. In music, neural activity synchronizes with the beat, the most prominent isochronous pulse in music to which listeners sway their bodies or tap their feet (*Tierney and Kraus, 2015*; *Nozaradan et al., 2012*; *Large and Snyder, 2009*; *Doelling and Poeppel, 2015*). Listeners show a strong behavioral preference for music with beat rates around 2 Hz (here, we use the term *tempo* to refer to the beat rate). The preference for 2 Hz coincides with the modal tempo of Western pop music (*Moelants, 2002*) and the most prominent frequency of natural adult body movements (*MacDougall and Moore, 2005*). Indeed, previous research showed that listeners perceive rhythmic sequences at beat rates around 1–2 Hz especially accurately when they are able to track the beat by moving their bodies (*Zalta*

**eLife digest** When we listen to a melody, the activity of our neurons synchronizes to the music: in fact, it is likely that the closer the match, the better we can perceive the piece. However, it remains unclear exactly which musical features our brain cells synchronize to. Previous studies, which have often used 'simplified' music, have highlighted that the amplitude envelope (how the intensity of the sounds changes over time) could be involved in this phenomenon, alongside factors such as musical training, attention, familiarity with the piece or even enjoyment. Whether differences in neural synchronization could explain why musical tastes vary between people is also still a matter of debate.

In their study, Weineck et al. aim to better understand what drives neuronal synchronization to music. A technique known as electroencephalography was used to record brain activity in 37 volunteers listening to instrumental music whose tempo ranged from 60 to 240 beats per minute. The tunes varied across an array of features such as familiarity, enjoyment and how easy the beat was to perceive. Two different approaches were then used to calculate neural synchronization, which yielded converging results.

The analyses revealed that three types of factors were associated with a strong neural synchronization. First, amongst the various cadences, a tempo of 60-120 beats per minute elicited the strongest match with neuronal activity. Interestingly, this beat is commonly found in Western pop music, is usually preferred by listeners, and often matches spontaneous body rhythms such as walking pace. Second, synchronization was linked to variations in pitch and sound quality (known as 'spectral flux') rather than in the amplitude envelope. And finally, familiarity and perceived beat saliency – but not enjoyment or musical expertise – were connected to stronger synchronization.

These findings help to better understand how our brains allow us to perceive and connect with music. The work conducted by Weineck et al. should help other researchers to investigate this field; in particular, it shows how important it is to consider spectral flux rather than amplitude envelope in experiments that use actual music.

et al., 2020). Despite the perceptual and motor evidence, studies looking at tempo-dependence of neural synchronization are scarce (*Doelling and Poeppel, 2015*; *Nicolaou et al., 2017*) and we are not aware of any human EEG study using naturalistic polyphonic musical stimuli that were manipulated in the tempo domain.

In the current study, we aimed to test whether the preference for music with beat rates around 2 Hz is reflected in the strength of neural synchronization by examining neural synchronization across a relatively wide and finely spaced range of musical tempi (1–4 Hz, corresponding to the neural δ band). In addition, a number of different musical, behavioral, and perceptual measures have been shown to modulate neural synchronization and influence music perception, including complexity, familiarity, repetition of the music, musical training of the listener, and attention to the stimulus (*Kumagai et al., 2018*; *Madsen et al., 2019*; *Doelling and Poeppel, 2015*). Thus, we investigated the effects of enjoyment, familiarity and the ease of beat perception on neural synchronization.

Most studies assessing neural synchronization to music have examined synchronization to either the stimulus amplitude envelope, which quantifies intensity fluctuations over time (*Doelling and Poeppel, 2015*; *Kaneshiro et al., 2020*; *Wollman et al., 2020*), or 'higher order' musical features such as surprise and expectation (*Di Liberto et al., 2020*). This mimics approaches used for studying neural synchronization to speech, where neural activity has been shown to synchronize with the amplitude envelope (*Peelle and Davis, 2012*), which roughly corresponds to syllabic fluctuations (*Doelling et al., 2014*), as well as to 'higher order' semantic information (*Broderick et al., 2019*). Notably, most studies that have examined neural synchronization to musical rhythm have used simplified musical stimuli, such as MIDI melodies (*Kumagai et al., 2018*) and monophonic melodies (*Di Liberto et al., 2020*), or rhythmic lines comprising clicks or sine tones (*Nozaradan et al., 2012*; *Nozaradan et al., 2011*; *Wollman et al., 2020*); only a few studies have focused on naturalistic, polyphonic music (*Tierney and Kraus, 2015*; *Madsen et al., 2019*; *Kaneshiro et al., 2020*; *Doelling and Poeppel, 2015*). 'Higher order' musical features are difficult to compute for naturalistic music, which is typically polyphonic and has complex spectro-temporal properties (*Zatorre et al., 2002*). However, amplitude-envelope synchronization is well documented: neural activity synchronizes to amplitude fluctuations

in music between 1 Hz and 8 Hz, and synchronization is especially strong for listeners with musical expertise (*Doelling and Poeppel, 2015*).

Because of the complex nature of natural polyphonic music, we hypothesized that amplitude envelope might not be the only or most dominant feature to which neural activity could synchronize (*Müller, 2015*). Thus, the current study investigated neural responses to different musical features that evolve over time and capture different aspects of the stimulus dynamics. Here, we use the term *musical feature* to refer to time-varying aspects of music that fluctuate on time scales corresponding roughly to the neural δ band, as opposed to elements of music such as key, harmony or syncopation. We examined amplitude envelope, the first derivative of the amplitude envelope (usually more sensitive to sound onsets than the amplitude envelope), beat times, and *spectral flux*, which describes spectral changes of the signal on a frame-to-frame basis by computing the difference between the spectral vectors of subsequent frames (*Müller, 2015*). One potential advantage of spectral flux over the envelope or its derivative is that spectral flux is sensitive to rhythmic information that is communicated by changes in pitch even when they are not accompanied by changes in amplitude. Critically, temporal and spectral information jointly influence the perceived accent structure in music, which provides information about beat locations (*Pfordresher, 2003*; *Ellis and Jones, 2009*; *Jones, 1993*).

The current study investigated neural synchronization to natural music by using two different analysis approaches: Reliable Components Analysis (RCA) (*Kaneshiro et al., 2020*) and temporal response functions (TRFs) (*Di Liberto et al., 2020*). A theoretically important distinction here is whether neural synchronization observed using these techniques reflects phase-locked, unidirectional coupling between a stimulus rhythm and activity generated by a neural oscillator (*Lakatos et al., 2019*) versus the convolution of a stimulus with the neural activity evoked by that stimulus (*Zuk et al., 2021*). TRF analyses involve modeling neural activity as a linear convolution between a stimulus and relatively broad-band neural activity (e.g. 1–15 Hz or 1–30 Hz; *Crosse et al., 2016*; *Crosse et al., 2021*); as such, there is a natural tendency for papers applying TRFs to interpret neural synchronization through the lens of convolution (although there are plenty of exceptions to this e.g. *Crosse et al., 2015*; *Di Liberto et al., 2015*). RCA-based analyses usually calculate correlation or coherence between a stimulus and relatively narrow-band activity, and in turn interpret neural synchronization as reflecting entrainment of a narrow-band neural oscillation to a stimulus rhythm (*Doelling and Poeppel, 2015*; *Assaneo et al., 2019*). Ultimately, understanding under what circumstances and using what techniques the neural synchronization we observe arises from either of these physiological mechanisms is an important scientific question (*Doelling et al., 2019*; *Doelling and Assaneo, 2021*; *van Bree et al., 2022*). However, doing so is not within the scope of the present study, and we prefer to remain agnostic to the potential generator of synchronized neural activity. Here, we refer to and discuss 'entrainment in the broad sense' (*Obleser and Kayser, 2019*) without making assumptions about *how* neural synchronization arises, and we will moreover show that these two classes of analyses techniques strongly agree with each other.

We aimed to answer four questions. (1) Does neural synchronization to natural music depend on tempo? (2) Which musical feature shows the strongest neural synchronization during natural music listening? (3) How compatible are RCA- and TRF-based methods at quantifying neural synchronization to natural music? (4) How do enjoyment, familiarity, and ease of beat perception affect neural synchronization? To answer these research questions, we recorded electroencephalography (EEG) data while participants listened to instrumental music presented at different tempi (1–4 Hz). Strongest neural synchronization was observed in response to the spectral flux of music, for tempi between 1 and 2 Hz, to familiar songs, and to songs with an easy-to-perceive beat.

## Results

Scalp EEG activity of 37 human participants was measured while they listened to instrumental segments of natural music from different genres (*Appendix 1—table 1*). Music segments were presented at thirteen parametrically varied tempi (1–4 Hz in 0.25 Hz steps; see Materials and methods). We assessed neural synchronization to four different musical features: amplitude envelope, first derivative of the amplitude envelope, beat times, and spectral flux. Neural synchronization was quantified using two different analysis pipelines and compared: (1) RCA combined with time- and frequency-domain analyses (*Kaneshiro et al., 2020*), and (2) TRFs (*Crosse et al., 2016*). As different behavioral and perceptual measures have been shown to influence neural synchronization to music (*Madsen et al., 2019*;

*Cameron et al., 2019*), we investigated the effects of enjoyment, familiarity, and the ease with which a beat was perceived (*Figure 1A*). To be able to use a large variety of musical stimuli on the group level, and to decrease any effects that may have arisen from individual stimuli occurring at certain tempi but not others, participants were divided into four subgroups that listened to different pools of stimuli (for more details please see Materials and methods). The subgroups' stimulus pools overlapped, but the individual song stimuli were presented at different tempi for each subgroup.

## Musical features

We examined neural synchronization to the time courses of four different musical features (*Figure 1B*). First, we quantified energy fluctuations over time as the gammatone-filtered amplitude envelope (we report analyses on the full-band envelope in *Figure 2—figure supplement 1* and *Figure 3—figure supplement 1*). Second, we computed the half-wave-rectified first derivative of the amplitude envelope, which is typically considered to be sensitive to the presence of onsets in the stimulus (*Bello et al., 2005*). Third, a percussionist drummed along with the musical segments to define beat times, which were here treated in a binary manner. Fourth, a spectral novelty function, referred to as spectral flux (*Müller, 2015*), was computed to capture changes in frequency content (as opposed to amplitude fluctuations) over time. In contrast to the first derivative, the spectral flux is better able to identify note onsets that are characterized by changes in spectral content (pitch or timbre), even if the energy level remains the same. To ensure that each musical feature possessed acoustic cues to the stimulation-tempo manipulation, we computed a fast Fourier transform (FFT) on the musical-feature time courses separately for each stimulation-tempo condition; the mean amplitude spectra are plotted in *Figure 1C*.

Overall, amplitude peaks were observed at the intended stimulation tempo and at the harmonic rates for all stimulus features.

In order to assess the degree to which the different musical features might have been redundant, we calculated mutual information (MI) for all possible pairwise feature combinations and compared MI values to surrogate distributions calculated separately for each feature pair (*Figure 1D and E*). MI quantifies the amount of information gained about one random variable by observing a second variable (*Cover and Thomas, 2005*). MI values were analyzed using separate three-way ANOVAs (MI data vs. MI surrogate ×Tempo × Subgroup) for each musical feature.

Spectral flux shared significant information with all other musical features; significant MI (relative to surrogate) was found between amplitude envelope and spectral flux ($F(1,102)=24.68$, $p_{FDR} = 1.01e-5$, $\eta^2=0.18$), derivative and spectral flux ($F(1,102)=82.3$, $p_{FDR} = 1.92e-13$, $\eta^2=0.45$) and beat times and spectral flux ($F(1,102)=23.05$, $p_{FDR} = 1.3e-5$, $\eta^2=0.13$). This demonstrates that spectral flux captures information from all three other musical features, and as such, we expected that spectral flux would be associated with strongest neural synchronization. Unsurprisingly, there was also significant shared information between the amplitude envelope and first derivative ($F(1,102)=14.11$, $p_{FDR} = 4.67e-4$, $\eta^2=0.09$); other comparisons: ($F_{env-beat}(1,102)=8.44$, $p_{FDR} = 0.006$, $\eta^2=0.07$; $F_{der-beat}(1,102)=6.06$, $p_{FDR} = 0.016$, $\eta^2=0.05$).

There was a main effect of Tempo on MI shared between the amplitude envelope and derivative ($F(12,91)=4$, $p_{FDR} = 2e-4$, $\eta^2=0.32$) and the spectral flux and beat times ($F(12,91)=5.48$, $p_{FDR} = 4.35e-6$, $\eta^2=0.37$) (*Figure 1—figure supplement 1*). This is likely due to the presence of slightly different songs in the different tempo conditions, as the effect of tempo on MI was unsystematic for both feature pairs (see Materials and methods and *Appendix 1—table 1*). MI for the remaining feature pairs did not differ significantly across tempi.

No significant differences in MI were observed between subgroups, despite the subgroups hearing slightly different pools of musical stimuli: ($F_{env-der}(3,100)=0.71$, $p_{FDR} = 0.94$, $\eta^2=0.01$; $F_{env-beat}(3,100)=2.63$, $p_{FDR} = 0.33$, $\eta^2=0.07$; $F_{env-spec}(3,100)=0.3$, $p_{FDR} = 0.94$, $\eta^2=0.01$; $F_{der-beat}(3,100)=0.43$, $p_{FDR} = 0.94$, $\eta^2=0.01$; $F_{der-spec}(3,100)=0.46$, $p_{FDR} = 0.94$, $\eta^2=0.01$; $F_{beat-spec}(3,100)=0.13$, $p_{FDR} = 0.94$, $\eta^2=0.002$).

## Neural synchronization was strongest in response to slow music

Neural synchronization to music was investigated using two converging analysis pipelines based on (1) RCA followed by time- (stimulus-response correlation, SRCorr) and frequency- (stimulus-response coherence, SRCoh) domain analysis and (2) TRFs.

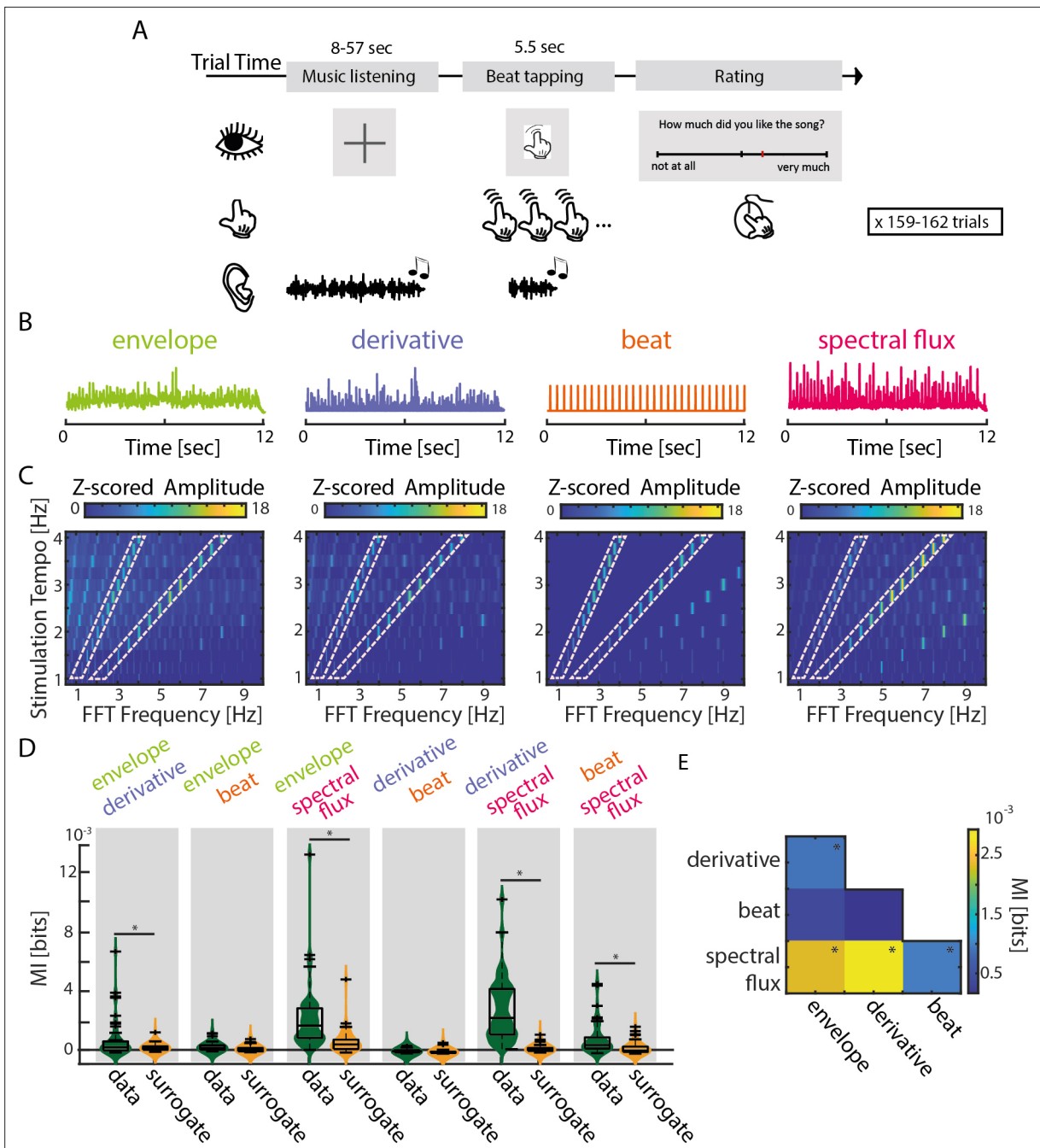

**Figure 1.** Experimental design and musical features. (**A**) Schematic of the experimental procedure. Each trial consisted of the presentation of one music segment, during which participants were instructed to listen attentively without moving. After a 1 s silence, the last 5.5 s of the music segment was repeated while participants tapped their finger along with the beat. At the end of each trial, participants rated their enjoyment and familiarity of the music segment, as well as the ease with which they were able to tap to the beat (Translated English example in Figure: "How much did you like the song?" rated from "not at all" to "very much"). (**B**) Exemplary traces of the four musical features of one music segment. (**C**) Z-scored mean amplitude spectrum of all 4 musical features. Light orange dashed boxes highlight when the FFT Frequency corresponds to the stimulation tempo or first harmonic. (**D**) Mutual information (MI) for all possible feature combinations (green) compared to a surrogate distribution (yellow, three-way ANOVA, *$p_{FDR}$ <0.001, rest: $p_{FDR}$ <0.05). Boxplots indicate the median, the 25th and 75th percentiles (n=52). (**E**) MI scores between all possible feature combinations (*$p_{FDR}$ <0.001, rest: $p_{FDR}$ <0.05).

The online version of this article includes the following source data and figure supplement(s) for figure 1:

**Source data 1.** Source data for visualizing and analyzing the musical features.

**Figure supplement 1.** Shared Mutual Information (MI) between musical features across tempo conditions.

**Figure supplement 2.** Tempo manipulations of original music segments.

First, an RCA-based analysis approach was used to assess tempo effects on neural synchronization to music (*Figure 2*, *Figure 2—figure supplement 1*). RCA involves estimating a spatial filter that maximizes correlation across data sets from multiple participants (for more details see Materials and methods) (*Kaneshiro et al., 2020*; *Parra et al., 2018*). The resulting time course data from a single reliable component can then be assessed in terms of its correlation in the time domain (SRCorr) or coherence in the frequency domain (SRCoh) with different musical feature time courses. Our analyses focused on the first reliable component, which exhibited an auditory topography (*Figure 2A*). To control for inherent tempo-dependent effects that could influence our results (such as higher power or variance at lower frequencies, that is 1/f), SRCorr and SRCoh values were normalized by a surrogate distribution. This way the temporal alignment between the stimulus and neural time course was destroyed, but the spectrotemporal composition of each signal was preserved. The surrogate distribution was obtained by randomly circularly shifting the neural time course in relation to the musical features per tempo condition and stimulation subgroup for 50 iterations (*Zuk et al., 2021*). Subsequently, the 'raw' SRCorr or SRCoh values were z-scored by subtracting the mean and dividing by the standard deviation of the surrogate distribution.

The resulting z-scored SRCorrs were significantly tempo-dependent for the amplitude envelope and the spectral flux (repeated-measure ANOVAs with Greenhouse-Geiser correction where required: $F_{env}(12,429)=2.5$, $p_{GG} = 0.015$, $\eta^2=0.07$; $F_{der}(12,429)=1.67$, p=0.07, $\eta^2=0.05$; $F_{beat}(12,429)=0.94$, p=0.5, $\eta^2=0.03$; $F_{spec}(12,429)=2.92$, $p_{GG} = 6.88e-4$, $\eta^2=0.08$). Highest correlations were found at slower tempi (~1–2 Hz).

No significant differences were observed across subgroups ($F_{env}(3,30)=1.13$, $p_{FDR=}0.55$, $\eta^2=0.1$; $F_{der}(3,30)=0.72$, $p_{FDR} = 0.55$, $\eta^2=0.07$; $F_{beat}(3,30)=0.85$, $p_{FDR} = 0.55$, $\eta^2=0.08$; $F_{spec}(3,30)=0.9$, $p_{FDR} = 0.55$, $\eta^2=0.08$). The results for the z-scored SRCorr were qualitatively similar to the 'raw' SRCorr with biggest differences for the beat feature.

In the frequency domain, z-scored SRCoh (*Figure 2D–G*) showed clear peaks at the stimulation tempo and harmonics. Overall, SRCoh was stronger at the first harmonic of the stimulation tempo than at the stimulation tempo itself, regardless of the musical feature (*Figure 2I*, paired-sample t-test, envelope: $t(12)=-5.16$, $p_{FDR} = 0.001$, $r_e = 0.73$; derivative: $t(12)=-5.11$, $p_{FDR} = 0.001$, $r_e = 0.72$; beat: $t(12)=-4.13$, $p_{FDR} = 0.004$, $r_e = 0.64$; spectral flux: $t(12)=-3.3$, $p_{FDR} = 0.01$, $r_e = 0.56$). The stimuli themselves mostly also contained highest FFT amplitudes at the first harmonic (*Figure 2J*, envelope: $t(12)=-6.81$, $p_{FDR} = 5.23e-5$, $r_e=0.81$; derivative: $t(12)=-6.88$, $p_{FDR} = 5.23e-5$, $r_e = 0.81$; spectral flux: $t(12)=-8.04$, $p_{FDR} = 2.98e-5$, $r_e = 0.85$), apart from the beat onsets (beat: $t(12)=6.27$, $p_{FDR} = 8.56–5$. $r_e = 0.79$).

For evaluating tempo-dependent effects, we averaged z-scored SRCoh across the stimulation tempo and first harmonic and submitted the average z-SRCoh values to repeated-measure ANOVAs for each musical feature. Z-SRCoh was highest for slow music, but this tempo dependence was only significant for the spectral flux and beat onsets ($F_{env}(12,429)=1.31$, p=0.21, $\eta^2=0.04$; $F_{der}(12,429)=1.71$, p=0.06, $\eta^2=0.05$; $F_{beat}(12,429)=2.07$, $p_{GG} = 0.04$, $\eta^2=0.06$; $F_{spec}(12,429)=2.82$, $p_{GG} = 0.006$, $\eta^2=0.08$). No significant differences for the SRCoh were observed across subgroups ($F_{env}(3,30)=0.93$, $p_{FDR=}0.58$, $\eta^2=0.09$; $F_{der}(3,30)=3.07$, $p_{FDR} = 0.17$, $\eta^2=0.24$; $F_{beat}(3,30)=2.26$, $p_{FDR} = 0.2$, $\eta^2=0.18$; $F_{spec}(3,30)=0.29$, $p_{FDR} = 0.83$, $\eta^2=0.03$). Individual data examples of the SRCorr and SRCoh can be found in *Figure 2—figure supplement 2*.

Second, TRFs were calculated for each stimulation tempo. A TRF-based approach is a linear-system identification technique that serves as a filter describing the mapping of stimulus features onto the neural response (forward model) (*Crosse et al., 2016*). Using linear convolution and ridge regression to avoid overfitting, the TRF was computed based on mapping each musical feature to 'training' EEG data. Using a leave-one-trial-out approach, the EEG response for the left-out trial was predicted based on the TRF and the stimulus feature of the same trial. The predicted EEG data were then correlated with the actual, unseen EEG data (we refer to this correlation value throughout as *TRF correlation*). We analyzed the two outputs of the TRF analysis: the filter at different time lags, which typically resembles evoked potentials, and the TRF correlations (*Figure 3*, *Figure 3—figure supplement 1*).

Again, strongest neural synchronization (here quantified as Pearson correlation coefficient between the predicted and actual EEG data) was observed for slower music (*Figure 3A*). After z-scoring the TRF correlations with respect to the surrogate distributions, as described for the SRcorr and SRCoh measures, repeated-measures ANOVAs showed that significant effects of Tempo were observed for all

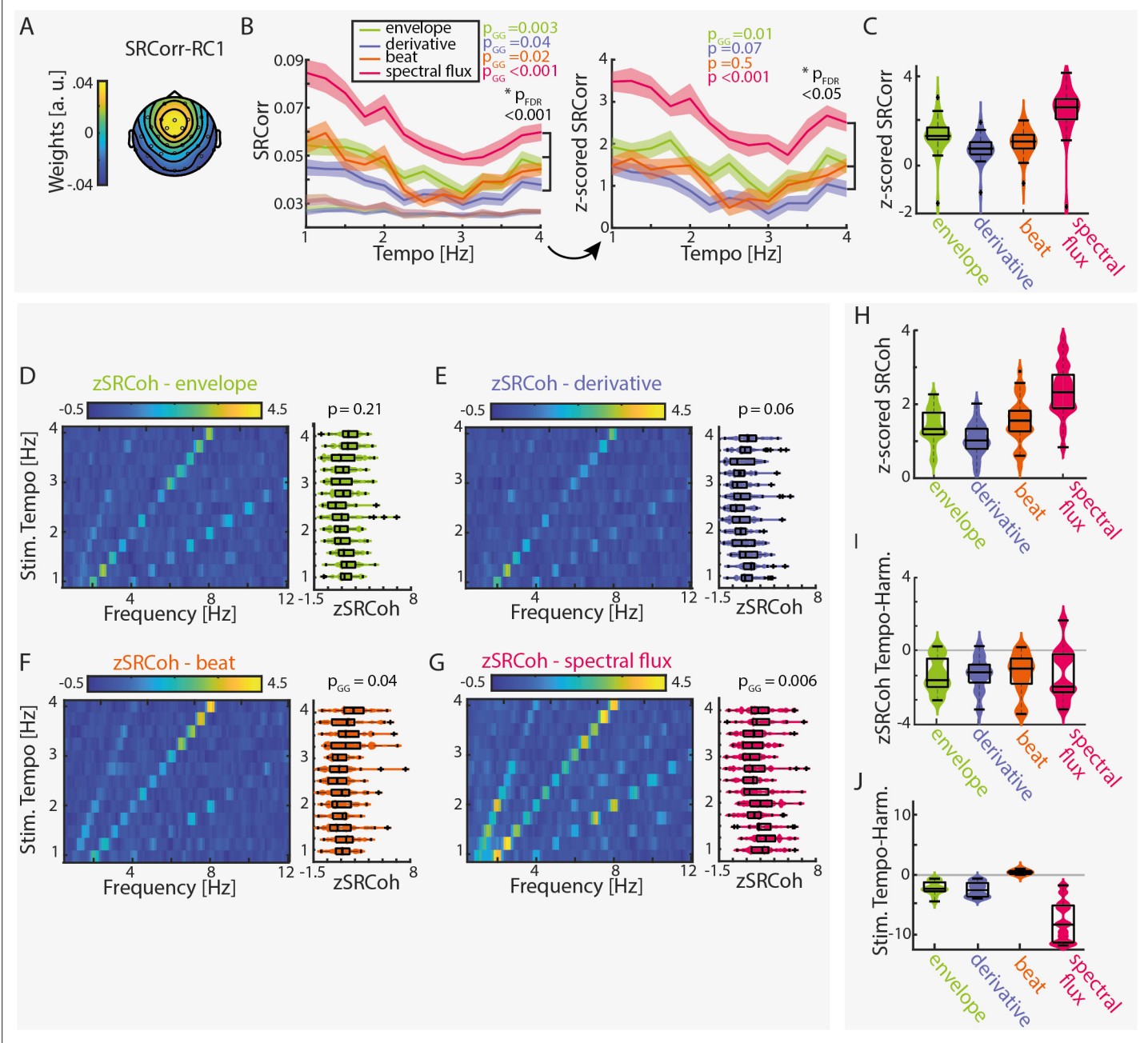

**Figure 2.** Stimulus–response correlation and stimulus–response coherence are tempo dependent for all musical features. (**A**) Projected topography of the first reliable component (RC1). (**B**) Average SRCorr of the aligned neural response and surrogate distribution (grey) across tempi for each musical feature (left) and the z-scored SRCorr based on a surrogate distribution (right) (± SEM; shaded area). Highest correlations were found at slow tempi (repeated-measure ANOVA, Greenhouse-Geiser correction where applicable). The slopes of regression models were used to compare the tempo-specificity between musical features. (**C**) Mean SRCorr across musical features. Highest correlations were found in response to spectral flux with significant differences between all possible feature combinations, $p_{FDR} < 0.001$, except between the envelope or derivative and beat onsets, $p_{FDR} < 0.01$ (n=34, repeated-measure ANOVA, Tukey's test, median, 25th and 75th percentiles). Z-scored SRCoh in response to the (**D**) amplitude envelope, (**E**) first derivative, (**F**) beat onsets and (**G**) spectral flux. Each panel depicts the SRCoh as colorplot (left) and the pooled SRCoh values at the stimulation tempo and first harmonic (right, n=34, median, 25th and 75th percentile). (**H**) Same as (**C**) for the SRCoh with significant differences between all possible feature combinations ($p_{FDR} < 0.001$) apart between the envelope and beat onsets. Coherence values were averaged over the stimulus tempo and first harmonic. (**I**) Mean differences of SRCoh values at the stimulation tempo and first harmonic (n=34, negative values: higher SRCoh at harmonic, positive values: higher SRCoh at stimulation tempo, paired-sample t-test, $p_{FDR} < 0.05$). (**J**) Same as (**I**) based on the FFT amplitudes ($p_{FDR} < 0.001$).

The online version of this article includes the following source data and figure supplement(s) for figure 2:

*Figure 2 continued on next page*

*Figure 2 continued*

**Source data 1.** Source data for the RCA-based measures stimulus-response correlation (SRCorr) and stimulus-response coherence (SRCoh).

**Source data 2.** Output of the RCA-based analysis of the first two stimulation subgroups (based on *Kaneshiro et al., 2020*).

**Source data 3.** Output of the RCA-based analysis of the last two stimulation subgroups (based on *Kaneshiro et al., 2020*).

**Figure supplement 1.** SRCorr and SRCoh in response to the full-band amplitude envelope and derivative.

**Figure supplement 2.** Individual data examples for the SRCorr and SRCoh.

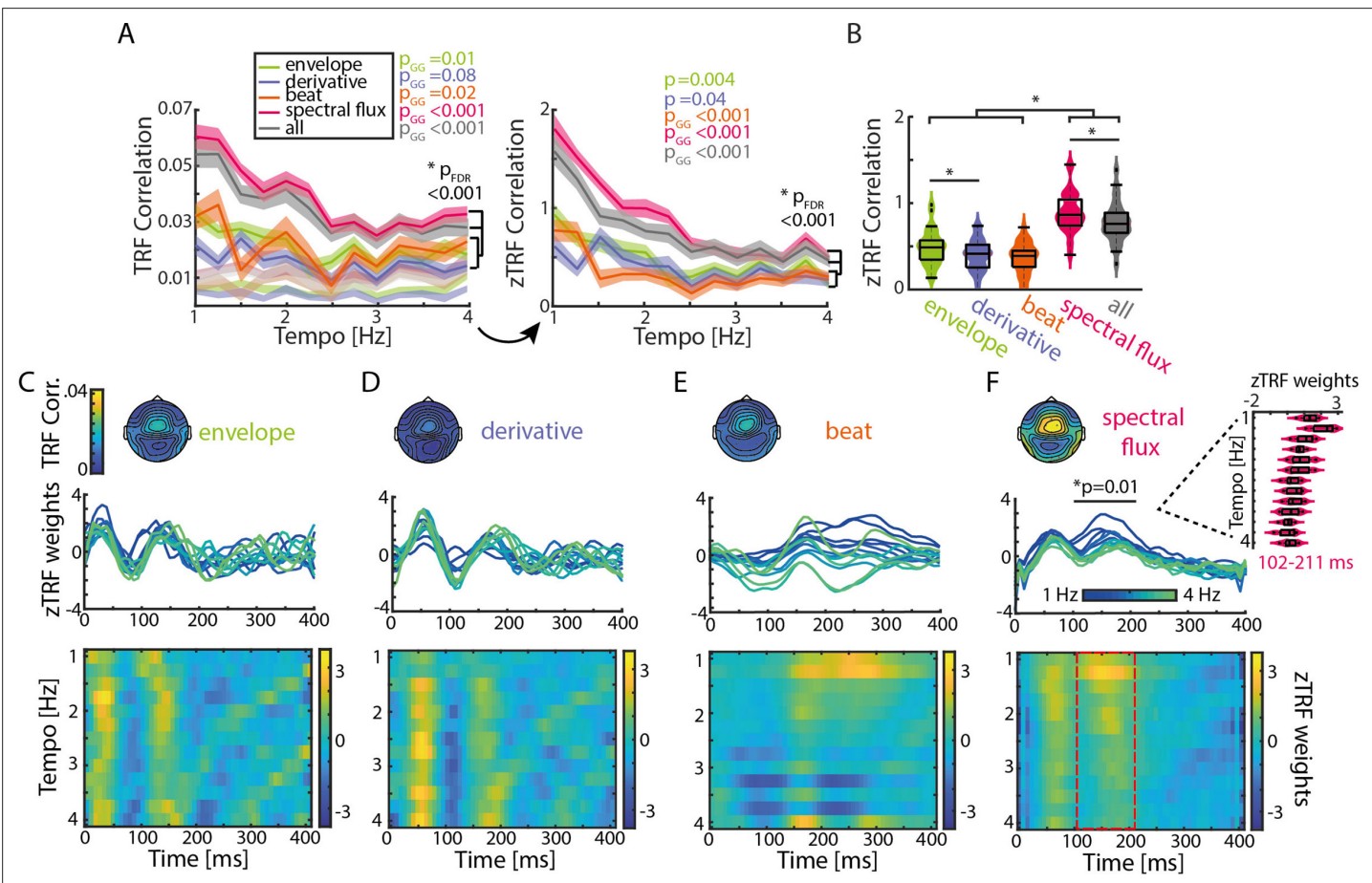

**Figure 3.** TRFs are tempo dependent. (**A**) Mean TRF (± SEM) correlations as a function of stimulation tempo per stimulus feature (p-values next to the legend correspond to a repeated-measure ANOVA across tempi for every musical feature and the p-value below to the slope comparison of a linear regression model). TRF correlations were highest for spectral flux and combined musical features for slow tempi. The TRF correlations were z-scored based on a surrogate distribution (right panel). (**B**) Violin plots of the TRF correlations across musical features. Boxplots illustrate the median, 25th and 75th percentiles (n=34). Significant pairwise musical feature comparisons were calculated using a repeated-measure ANOVA with follow-up Tukey's test, *$p_{FDR}$ <0.001. (**C**) Top panel: Topographies of the TRF correlations and TRF time lags (0–400ms) in response to the amplitude envelope. Each line depicts one stimulation tempo (13 tempi between 1 Hz, blue and 4 Hz, green). Lower panel: Colormap of the normalized TRF weights of the envelope in the same time window across stimulation tempi. (**D**) Same as (**C**) for the first derivative, (**E**) beat onsets and (**F**) spectral flux. Cluster-based permutation testing was used to identify significant tempo-specific time windows (red dashed box, p<0.05). Inset: Mean TRF weights in response to the spectral flux for time lags between 102 and 211ms (n=34, median, 25th and 75th percentile).

The online version of this article includes the following source data and figure supplement(s) for figure 3:

**Source data 1.** Source data of the TRF correlations and weights.

**Figure supplement 1.** TRFs in response to the full-band amplitude envelope and first derivative show similar patterns as the gammatone filtered musical features.

**Figure supplement 2.** Corrected TRF weights of the spectral flux after removing the effects of the other musical features.

**Figure supplement 3.** No differences in TRFs correlations between more vs. less modulated music.

musical features with z-TRF correlations being strongest at slower tempi (~1–2 Hz) ($F_{env}$(12,429)=2.47, p=0.004, $\eta^2$=0.07; $F_{der}$(12,429)=1.84, $p_{GG}$ = 0.04, $\eta^2$=0.05; $F_{beat}$(12,429)=3.81, $p_{GG}$ = 3.18e-4, $\eta^2$=0.11; $F_{spec}$(12,429)=12.87, $p_{GG}$ = 3.87e-13, $\eta^2$=0.29).

The original tempi of the music segments prior to being tempo manipulated fell mostly into the range spanning 1.25–2.5 Hz (*Figure 1—figure supplement 2A*). Thus, music that was presented at stimulation tempi in this range were shifted to a smaller degree than music presented at tempi outside of this range, and music presented at slow tempi tended to be shifted to a smaller degree than music presented at fast tempi (*Figure 1—figure supplement 2B*,C). Thus, we conducted a control analysis to show that there was no significant effect on z-TRF correlations of how much music stimuli were tempo shifted (2.25 Hz: F(2,96)=0.45, *P*=0.43; 1.5 Hz: F(2,24)=0.49, p=0.49; *Figure 3—figure supplement 3*; for more details see Materials and methods).

## Spectral flux drives strongest neural synchronization

As natural music is a complex, multi-layered auditory stimulus, we sought to explore neural synchronization to different musical features and to identify the stimulus feature or features that would drive strongest neural synchronization. Regardless of the dependent measure (RCA-SRCorr, RCA-SRCoh, TRF correlation), strongest neural synchronization was found in response to spectral flux (*Figures 2C, H and 3B*). In particular, significant differences (as quantified with a repeated-measure ANOVA followed by Tukey's test) were observed between the spectral flux and all other musical features based on z-scored SRCorr ($F_{SRCorr}$(3,132)=39.27, $p_{GG}$ = 1.2e-16, $\eta^2$=0.55), z-SRCoh ($F_{SRCoh}$(3,132)=26.27, $p_{GG}$ = 1.72e-12, $\eta^2$=0.45) and z-TRF correlations ($F_{TRF}$(4,165)=30.09, $p_{GG}$ = 1.21e-13, $\eta^2$=0.48).

As the TRF approach offers the possibility of running a multivariate analysis, all musical features were combined and the resulting z-scored TRF correlations were compared to the single-feature TRF correlations (*Figure 3B*). Although there was a significant increase in z-TRF correlations in comparison to the amplitude envelope (repeated-measure ANOVA with follow-up Tukey's test, $p_{FDR}$ = 1.66e-08), first derivative ($p_{FDR}$ = 1.66e-8), and beat onsets ($p_{FDR}$ = 1.66e-8), the spectral flux alone showed an advantage over the multi-featured TRF ($p_{FDR}$ = 6.74e-8). Next, we ran a multivariate TRF analysis combining amplitude envelope, first derivative, and beat onsets, and then subtracted the predicted EEG data from the actual EEG data (*Figure 3—figure supplement 2*). We calculated a TRF forward model using spectral flux to predict the EEG data residualized with respect to the multivariate predictor combining the remaining musical features. The resulting TRF weights were qualitatively similar to the model with spectral flux as the only predictor of the neural response. Thus, taking all stimulus features together is not a better descriptor of the neural response than the spectral flux alone, indicating together with the MI results from *Figure 1* that spectral flux is a more complete representation of the rhythmic structure of the music than the other musical features.

To test how strongly modulated TRF correlations were by tempo for each musical feature, a linear regression was fitted to single-participant z-TRF correlations as a function of tempo, and the slopes were compared across musical features (*Figure 3A*). Linear slopes were significantly higher for spectral flux and the multivariate model compared to the remaining three musical features (repeated-measure ANOVA with follow-up Tukey's test, envelope-spectral flux: $p_{FDR}$ = 2.8e-6; envelope – all: $p_{FDR}$ = 2.88e-4; derivative-spectral flux: $p_{FDR}$ = 7.47e-8; derivative – all: $p_{FDR}$ = 2.8e-6; beat-spectral flux: $p_{FDR}$ = 2.47e-8; beat – all: $p_{FDR}$ = 2.09e-5; spectral flux – all: $p_{FDR}$ = 0.01). The results for z-SRCorr were qualitatively similar except for the comparison between the envelope and spectral flux (envelope-spectral flux: $p_{FDR}$ = 0.12; derivative-spectral flux: $p_{FDR}$ = 0.04; beat-spectral flux: $p_{FDR}$ = 6e-4; *Figure 2B*).

Finally, we also examined the time courses of TRF weights (*Figure 3C–F*) for time lags between 0 and 400ms, and how they depended on tempo. Cluster-based permutation testing (1,000 repetitions) was used to identify time windows in which TRF weights differed across tempi for each musical feature (see Materials and methods for more details). Significant effects of tempo on TRF weights were observed for spectral flux between 102–211ms (p=0.01; *Figure 3F*). The tempo specificity was observable in the amplitudes of the TRF weights, which were largest for slower music (*Figure 3F*). The TRFs for the amplitude envelope and first derivative demonstrated similar patterns to each other, with strong deflections in time windows consistent with a canonical auditory P1–N1–P2 complex, but did not differ significantly between stimulation tempi (*Figure 3C–D*). Similarly, the full-band (Hilbert) amplitude envelope and the corresponding first derivative (*Figure 3—figure supplement 1*) displayed tempo-specific effects at time lags of 250–400ms (envelope, p=0.01) and 281–400ms (derivative,

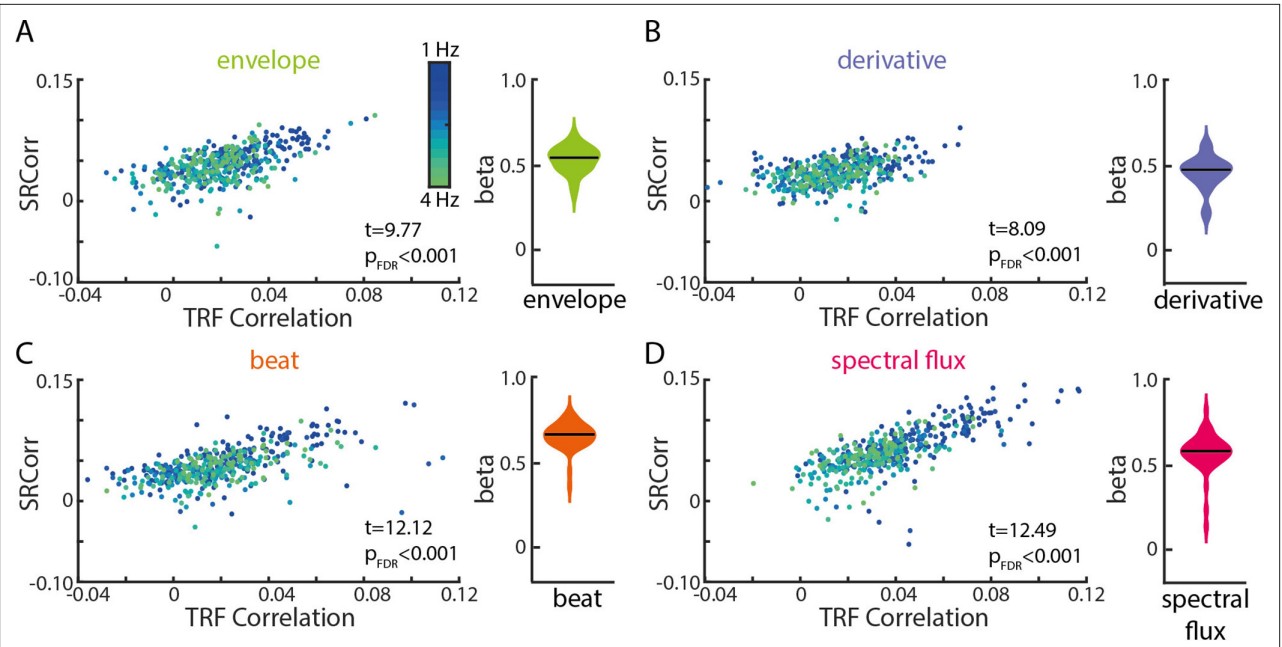

**Figure 4.** Significant relationships between SRCorr and TRF correlations for all musical features. (**A**) Linear-mixed effects models of the SRCorr (predictor variable) and TRF correlations (response variable) in response to the amplitude envelope. Each dot represents the mean correlation of one participant (n=34) at one stimulation tempo (n=13) (=grouping variables; blue, 1 Hz-green, 4 Hz). Violin plots illustrate fixed effects coefficients (β). (**B**)-(**D**) same as (**A**) for the first derivative, beat onsets and spectral flux. For all musical features, the fixed effects were significant.

The online version of this article includes the following source data and figure supplement(s) for figure 4:

**Source data 1.** Source data for comparing the results of the TRF and RCA-based measures.

**Figure supplement 1.** Significant relationships between SRCoh and TRF correlations for all musical features at the stimulation tempo and first harmonic.

p=0.02). Visual inspection suggested that TRF differences for these musical features were related to latency, as opposed to amplitude (*Figure 3—figure supplement 1E-F,I-J*). Therefore, we identified the latencies of the TRF-weight time courses within the time window of P3 and fitted a piece-wise linear regression to those mean latency values per musical feature (*Figure 3—figure supplement 1G, K*). In particular, TRF latency in the P3 time window decreased over the stimulation tempo conditions from 1 to 2.5 Hz and from 2.75 to 4 Hz for both stimulus features (derivative: $T_{1-2.5Hz}$=-1.08, p=0.33, $R^2$=0.03; $T_{2.75-4Hz}$=-2.2, p=0.09, $R^2$=0.43), but this was only significant for the envelope ($T_{1-2.5Hz}$=-6.1, p=0.002, $R^2$=0.86; $T_{2.75-4Hz}$=-5.66, p=0.005, $R^2$=0.86).

### Results of TRF and SRCorr/SRCoh converge

So far, we demonstrated that both RCA- and TRF-based measures of neural synchronization led to similar results at the group level, and reveal strongest neural synchronization to spectral flux and at slow tempi. Next, we wanted to quantify the relationship between the SRCorr/SRCoh and TRF correlations across individuals (*Figure 4*, *Figure 4—figure supplement 1*). This could have implications for the interpretation of studies focusing only on one method. To test this relationship, we predicted TRF correlations from SRCorr or SRCoh values (fixed effect) in separate linear mixed-effects models with Participant and Tempo as random effects (grouping variables). For all further analyses, we used the 'raw' (non-z-scored) values for all dependent measures, as they yielded in the previous analysis (*Figures 2 and 3*) qualitatively similar results to the z-scored values. Each musical feature was modeled independently.

For all four musical features, SRCorr significantly predicted TRF correlations ($t_{env}$(440) = 9.77, $β_{env}$=0.53, $p_{FDR}$ <1e-15, $R^2$=0.51; $t_{der}$(440) = 8.09, $β_{der}$=0.46, $p_{FDR}$ = 5.77e-14, $R^2$=0.28; $t_{beat}$(440) = 12.12, $β_{beat}$=0.67, $p_{FDR}$ <1e-15, $R^2$=0.61; $t_{spec}$(440) = 12.49, $β_{spec}$=0.56, $p_{FDR}$ = 1e-15, $R^2$=0.76). The strongest correlations between neural synchronization measures were found for the beat onsets and spectral flux of music (*Figure 4C and D*).

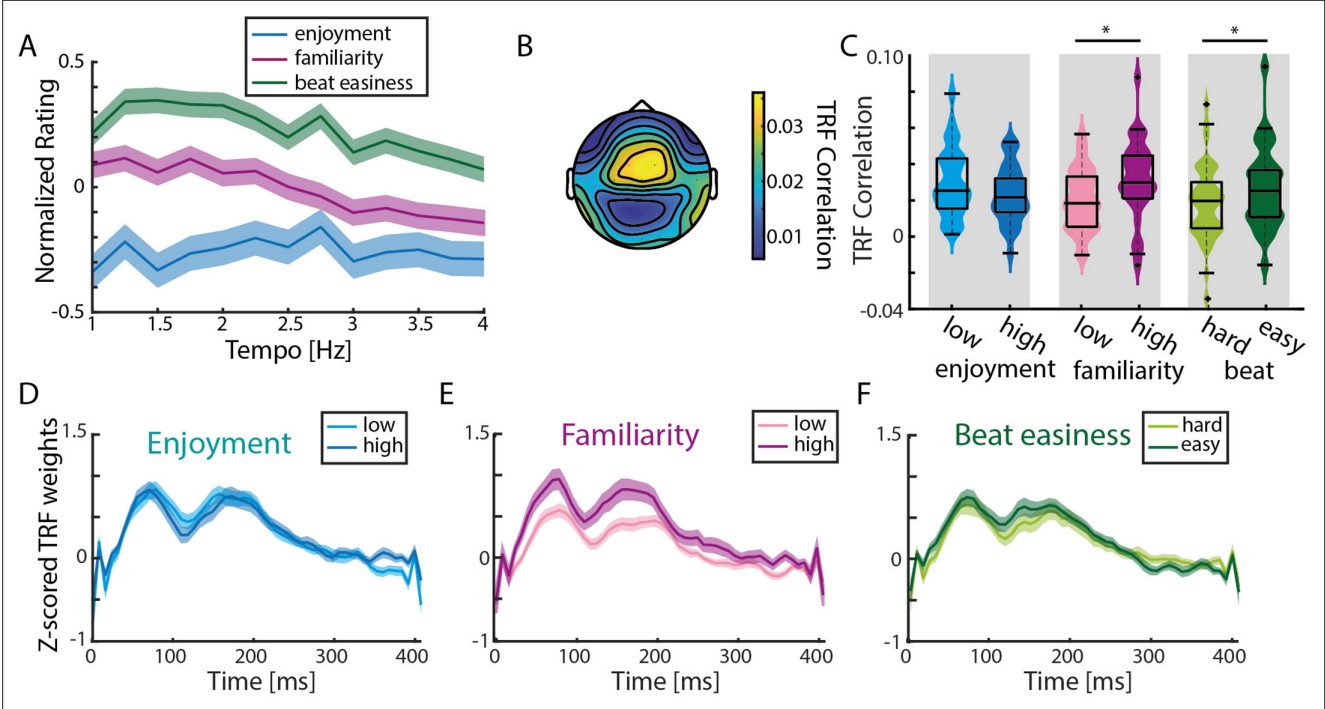

**Figure 5.** TRF correlations are highest in response to familiar songs. (**A**) Normalized (to the maximum value per rating/participant), averaged behavioral ratings of enjoyment, familiarity and easiness to tap to the beat (± SEM). No significant differences across tempo conditions were observed (repeated-measure ANOVA with Greenhouse-Geiser correction). (**B**) Mean TRF correlations topography across all ratings (based on the analysis of 15 trials with highest and lowest ratings per behavioral measure). (**C**) Violin plots of TRF correlations comparing low vs. highly enjoyed, low vs. highly familiar, and subjectively difficult vs. easy beat trials. Strongest TRF correlations were found in response to familiar music and music with an easy-to-perceive beat (n=34, paired-sample t-test, *$p_{FDR}$ <0.05). Boxplots indicate median, 25th and 75th percentile. (**D**) Mean TRFs (± SEM) for time lags between 0–400ms of more and less enjoyable music songs. (**E**)-(**F**) Same as (**D**) for trials with low vs. high familiarity and difficult vs. easy beat ratings.

The online version of this article includes the following source data and figure supplement(s) for figure 5:

**Source data 1.** Source data of the behavioral ratings and TRF correlations.

**Figure supplement 1.** Significant differences of FFT amplitudes at stimulus-relevant frequencies between differently rated trials.

**Figure supplement 2.** Musical training did not have an effect on TRF correlations regardless of the musical feature.

**Figure supplement 3.** Music tapping rate across participants.

In the frequency domain, we examined the SRCoh values at the stimulation tempo and first harmonic separately (*Figure 4—figure supplement 1*). SRCoh values at both the intended stimulation tempo and the first harmonic significantly predicted TRF correlations for all musical features. For all musical features, the first harmonic was a better predictor of TRF correlations than the intended stimulation tempo except for the beat onsets (intended tempo: $t_{env}(440) = 4.78$, $\beta_{env}=0.17$, $p_{FDR} = 3.15e-6$, $R^2=0.34$; $t_{der}(440) = 3.06$, $\beta_{der}=0.1$, $p_{FDR} = 0.002$, $R^2=0.13$; $t_{beat}(440) = 8.12$, $\beta_{beat}=0.28$, $p_{FDR} = 1.95e-14$, $R^2=0.5$; $t_{spec}(440) = 3.42$, $\beta_{spec}=0.09$, $p_{FDR} = 7.9e-4$, $R^2=0.64$; first harmonic: $t_{env}(440) = 6.17$, $\beta_{env}=0.09$, $p_{FDR} = 3.07e-9$, $R^2=0.33$; $t_{der}(440) = 4.98$, $\beta_{der}=0.09$, $p_{FDR} = 1.43e-6$, $R^2=0.16$; $t_{beat}(440) = 8.79$, $\beta_{beat}=0.2$, $p_{FDR} <1e-15$, $R^2=0.51$; $t_{spec}(440) = 6.87$, $\beta_{spec}=0.09$, $p_{FDR} = 5.82e-11$, $R^2=0.64$). Overall, these results suggest that, despite their analytical differences as well as common differences in interpretation, TRF and RCA–SRCorr/RCA-SRCoh seem to pick up on similar features of the neural response, but may potentially strengthen each other's explanatory power when used together.

## Familiar songs and songs with an easy-to-tap beat drive strongest neural synchronization

Next, we tested whether neural synchronization to music depended on (1) how much the song was enjoyed, (2) the familiarity of the song, and (3) how easy it was to tap the beat of the song; each of these characteristics was rated on a scale ranging between –100 and +100. We hypothesized that

difficulty to perceive and tap to the beat in particular would be associated with weaker neural synchronization. Ratings on all three dimensions are shown in *Figure 5A*. To evaluate the effects of tempo on the individuals' ratings, separate repeated-measure ANOVAs were conducted for each behavioral rating. All behavioral ratings were unaffected by tempo (enjoyment: F(12,429)=0.58, p=0.85, $\eta^2$=0.02; familiarity: F(12,429)=1.44, $p_{GG}$ = 0.18, $\eta^2$=0.04; ease of beat tapping: F(12,429)=1.62, P=0.08, $\eta^2$=0.05).

To assess the effects of familiarity, enjoyment, and beat-tapping ease on neural synchronization, TRFs in response to spectral flux were calculated for the 15 trials with the highest and the 15 trials with the lowest ratings per participant per behavioral rating (*Figure 5B–F*). TRF correlations were not significantly different for less enjoyed compared to more enjoyed music (paired-sample t-test, t(33)=1.91, $p_{FDR}$ = 0.06, $r_e$ = 0.36; *Figure 5C*). In contrast, significantly higher TRF correlations were observed for familiar vs. unfamiliar songs (t(33)=-2.57, $p_{FDR}$ = 0.03, $r_e$ = 0.46), and for songs with an easier-to-perceive beat (t(33)=-2.43, $p_{FDR}$ = 0.03, $r_e$ = 0.44). These results were reflected in the TRFs at time lags between 0 and 400ms (*Figure 5D–F*). We wanted to test whether these TRF differences may have been attributable to acoustic features, such as the beat salience of the musical stimuli, which could have an effect on both behavioral ratings and TRFs. Thus, we computed single-trial FFTs on the spectral flux of the 15 highest vs. lowest rated trials (*Figure 5—figure supplement 1*). Pairwise comparisons revealed higher stimulus-related FFT peaks for more enjoyed music (t-test, t(33)=-2.79, $p_{FDR}$ = 0.01, $r_e$ = 0.49), less familiar music (t(33)=2.73, $p_{FDR}$ = 0.01, $r_e$ = 0.49) and easier-to-perceive beats (t(33)=-3.33, $p_{FDR}$ = 0.01, $r_e$ = 0.56).

Next, we wanted to entertain the possibility that musical expertise could modulate neural synchronization to music (*Doelling and Poeppel, 2015*). We used the Goldsmith's Musical Sophistication Index (Gold-MSI) to quantify musical 'sophistication' (referring not only to the years of musical training, but also e. g. musical engagement or self-reported perceptual abilities *Müllensiefen et al., 2014*), which we then correlated with neural synchronization. No significant correlations were observed between musical sophistication and TRF correlations (Pearson correlation, envelope: *R*=−0.21, $p_{FDR}$ = 0.32; derivative: *R*=−0.24, $p_{FDR}$ = 0.31; beats: *R*=−0.04, $p_{FDR}$ = 0.81; spectral flux: *R*=−0.34, $p_{FDR}$ = 0.2; *Figure 5—figure supplement 2*).

## Discussion

We investigated neural synchronization to naturalistic, polyphonic music presented at different tempi. The music stimuli varied along a number of dimensions in idiosyncratic ways, including the familiarity and enjoyment of the music, and the ease with which the beat was perceived. The current study demonstrates that neural synchronization is strongest to (1) music with beat rates between 1 and 2 Hz, (2) spectral flux of music, and (3) familiar music and music with an easy-to-perceive beat. In addition, (4) analysis approaches based on TRF and RCA revealed converging results.

### Neural synchronization was strongest to music with beat rates in the 1–2 Hz range

Strongest neural synchronization was found in response to stimulation tempi between 1 and 2 Hz in terms of SRCorr (*Figure 2B*), TRF correlations (*Figure 3A*), and TRF weights (*Figure 3C–F*). Moreover, we observed a behavioral preference to tap to the beat in this frequency range, as the group preference for music tapping was at 1.55 Hz (*Figure 5—figure supplement 3*). Previous studies have shown a preference to listen to music with beat rates around 2 Hz (*Bauer et al., 2015*), which is moreover the modal beat rate in Western pop music (*Moelants, 2002*) and the rate at which the modulation spectrum of natural music peaks (*Ding et al., 2017*). Even in nonmusical contexts, spontaneous adult human locomotion is characterized by strong energy around 2 Hz (*MacDougall and Moore, 2005*). Moreover, when asked to rhythmically move their bodies at a comfortable rate, adults will spontaneously move at rates around 2 Hz (*McAuley et al., 2006*) regardless whether they use their hands or feet (*Rose et al., 2020*). Thus, there is a tight link between preferred rates of human body movement and preferred rates for the music we make and listen to that was moreover reflected in our neural data. This is perhaps not surprising, as musical rhythm perception activates motor areas of the brain, such as the basal ganglia and supplementary motor area (*Grahn and Brett, 2007*), and is further associated with increased auditory–motor functional connectivity (*Chen et al., 2008*). In turn, involving the

motor system in rhythm perception tasks improves temporal acuity (*Morillon et al., 2014*), but only for beat rates in the 1–2 Hz range (*Zalta et al., 2020*).

The tempo range within which we observed strongest synchronization partially coincides with the original tempo range of the music stimuli (*Figure 1—figure supplement 2*). A control analysis revealed that the amount of tempo manipulation (difference between original music tempo and tempo at which the music segment was presented to the participant) did not affect TRF correlations. Thus, we interpret our data as reflecting a neural preference for specific musical tempi rather than an effect of naturalness or the amount that we had to tempo shift the stimuli. However, since our experiment was not designed to answer this question, we were only able to conduct this analysis for two tempi, 2.25 Hz and 1.5 Hz (*Figure 3—figure supplement 3*), and thus are not able to rule out the influence of the magnitude of tempo manipulation on other tempo conditions.

In the frequency domain, SRCoh was strongest at the stimulation tempo and its harmonics (*Figure 2D–G,I*). In fact, highest coherence was observed at the first harmonic and not at the stimulation tempo itself (*Figure 2I*). This replicates previous work that also showed higher coherence (*Kaneshiro et al., 2020*) and spectral amplitude (*Tierney and Kraus, 2015*) at the first harmonic than at the musical beat rate. There are several potential reasons for this finding. One reason could be that the stimulation tempo that we defined for each musical stimulus was based on beat rate, but natural music can be subdivided into smaller units (e.g. notes) that can occur at faster time scales. A recent MEG study demonstrated inter-trial phase coherence for note rates up to 8 Hz (*Doelling and Poeppel, 2015*). Hence, the neural responses to the music stimuli in the current experiment were likely synchronized to not only the beat rate, but also faster elements such as notes. In line with this hypothesis, FFTs conducted on the stimulus features themselves showed higher amplitudes at the first harmonic than the stimulation tempo for all musical features except the beat onsets (*Figure 2J*). Moreover, there are other explanations for higher coherence at the first harmonic than at the beat rate. For example, the low-frequency beat-rate neural responses fall into a steeper part of the 1 /f slope, and as such may simply suffer from worse signal-to-noise ratio than their harmonics.

Regardless of the reason, since frequency-domain analyses separate the neural response into individual frequency-specific peaks, it is easy to interpret neural synchronization (SRCoh) or stimulus spectral amplitude at the beat rate and the note rate – or at the beat rate and its harmonics – as independent (*Keitel et al., 2021*). However, music is characterized by a nested, hierarchical rhythmic structure, and it is unlikely that neural synchronization at different metrical levels goes on independently and in parallel. One potential advantage of TRF-based analyses is that they operate on relatively wide-band data compared to Fourier-based approaches, and as such are more likely to preserve nested neural activity and perhaps less likely to lead to over- or misinterpretation of frequency-specific effects.

## Neural synchronization is driven by spectral flux

Neural synchronization was strongest in response to the spectral flux of music, regardless whether the analysis was based on TRFs or RCA. Similar to studies using speech stimuli, music studies typically use the amplitude envelope of the sound to characterize the stimulus rhythm (*Vanden Bosch der Nederlanden et al., 2020*; *Kumagai et al., 2018*; *Doelling and Poeppel, 2015*; *Decruy et al., 2019*; *Reetzke et al., 2021*). Although speech and music share features such as amplitude fluctuations over time and hierarchical grouping (*Patel, 2003*), there are differences in their spectro-temporal composition that make spectral information especially important for music perception. For example, while successful speech recognition requires 4–8 spectral channels, successful recognition of musical melodies requires at least 16 spectral channels (*Shannon, 2005*) – the flipside of this is that music is more difficult than speech to understand based only on amplitude-envelope information. Moreover, increasing spectral complexity of a music stimulus enhances neural synchronization (*Wollman et al., 2020*). Previous work on joint accent structure indicates that spectral information is an important contributor to beat perception (*Ellis and Jones, 2009*; *Pfordresher, 2003*). Thus, it was our hypothesis in designing the current study that a feature that incorporates spectral changes over time, as opposed to amplitude differences only, would better capture how neural activity entrains to musical rhythm.

Using TRF analysis, we found that not only was neural synchronization to spectral flux stronger than to any other musical feature, it was also stronger than the response to a multivariate predictor that

combined all musical features. For this reason, we calculated the shared information (MI) between each pair of musical features, and found that spectral flux shared significant information with all other musical features (*Figure 1*). Hence, spectral flux seems to capture information contained in, for example, the amplitude envelope, but also to contain unique information about rhythmic structure that cannot be gleaned from the other acoustic features (*Figure 3*).

One hurdle to performing any analysis of the coupling between neural activity and a stimulus time course is knowing ahead of time the feature or set of features that will well characterize the stimulus on a particular time scale given the nature of the research question. Indeed, there is no necessity that the feature that best drives neural synchronization will be the most obvious or prominent stimulus feature. Here, we treated feature comparison as an empirical question (*Di Liberto et al., 2015*), and found that spectral flux is a better predictor of neural activity than the amplitude envelope of music. Beyond this comparison though, the issue of feature selection also has important implications for comparisons of neural synchronization across, for example, different modalities.

For example, a recent study found that neuronal activity synchronizes less strongly to music than to speech *Zuk et al., 2021*; notably this paper focused on the amplitude envelope to characterize the rhythms of both stimulus types. However, our results show that neural synchronization is especially strong to the spectral content of music, and that spectral flux may be a better measure for capturing musical dynamics than the amplitude envelope (*Müller, 2015*). Imagine listening to a melody played in a *glissando* fashion on a violin. There might never be a clear onset that would be represented by the amplitude envelope – all of the rhythmic structure is communicated by spectral changes. Indeed, many automated tools for extracting the beat in music used in the musical information retrieval (MIR) literature rely on spectral flux information (*Olivera et al., 2010*). Also in the context of body movement, spectral flux has been associated with the type and temporal acuity of synchronization between the body and music at the beat rate (*Burger et al., 2018*) to a greater extent than other acoustic characterizations of musical rhythmic structure. As such, we found that spectral flux synchronized brain activity better than the amplitude envelope.

## Neural synchronization was strongest to familiar songs and songs with an easy beat

We found that the strength of neural synchronization depended on the familiarity of music and the ease with which a beat could be perceived (*Figure 5*). This is in line with previous studies showing stronger neural synchronization to familiar music (*Madsen et al., 2019*) and familiar sung utterances (*Vanden Bosch der Nederlanden et al., 2022*). Moreover, stronger synchronization for musicians than for nonmusicians has been interpreted as reflecting musicians' stronger expectations about musical structure. On the surface, these findings might appear to contradict work showing stronger responses to music that violated expectations in some way (*Kaneshiro et al., 2020*; *Di Liberto et al., 2020*). However, we believe these findings are compatible: familiar music would give rise to stronger expectations and stronger neural synchronization, and stronger expectations would give rise to stronger 'prediction error' when violated. In the current study, the musical stimuli never contained violations of any expectations, and so we observed stronger neural synchronization to familiar compared to unfamiliar music. There was also higher neural synchronization to music with subjectively 'easy-to-tap-to' beats. Overall, we interpret our results as indicating that stronger neural synchronization is evoked in response to music that is more predictable: familiar music and with easy-to-track beat structure.

Musical training did not affect the degree of neural synchronization in response to tempo-modulated music (*Figure 5—figure supplement 2*). This contrasts with previous music research showing that musicians' neural activity was entrained more strongly by music than non-musicians' (*Madsen et al., 2019*; *Doelling and Poeppel, 2015*; *Di Liberto et al., 2020*). There are several possible reasons for this discrepancy. One is that most studies that have observed differences between musicians and nonmusicians focused on classical music (*Doelling and Poeppel, 2015*; *Madsen et al., 2019*; *Di Liberto et al., 2020*), whereas we incorporated music stimuli with different instruments and from different genres (e.g. Rock, Pop, Techno, Western, Hip Hop, or Jazz). We suspect that musicians are more likely to be familiar with, in particular, classical music, and as we have shown that familiarity with the individual piece increases neural synchronization, these studies may have inadvertently confounded musical training with familiarity. Another potential reason for the lack of effects of musical training on neural synchronization in the current study could originate from the choice of utilizing

acoustic audio descriptors as opposed to 'higher order' musical features. However, 'higher order' features such as surprise or entropy that have been shown to be influenced by musical expertise (*Di Liberto et al., 2020*) are difficult to compute for natural, polyphonic music.

### TRF- and RCA-based measures show converging results

RCA and TRF approaches share their ability to characterize neural responses to single-trial, ongoing, naturalistic stimuli. As such, both techniques afford something that is challenging or impossible to accomplish with 'classic' ERP analysis. However, we made use of two techniques in parallel in order to leverage their unique advantages. RCA allows for frequency-domain analysis such as SRCoh, which can be useful for identifying neural synchronization specifically at the beat rate, for example. The frequency-specificity could serve as an advantage of the SRCoh over the TRF measures, where an EEG broadband signal was used. However, the RCA-based approaches *Kaneshiro et al., 2020* have been criticized because of their potential susceptibility to autocorrelation, which is argued to be minimized in the TRF approach (*Zuk et al., 2021*), which uses ridge regression to dampen fast oscillatory components (*Crosse et al., 2021*). However, by minimizing the effects of auto-correlation one concern could be that this could remove neural oscillations of interest as well. TRFs also offer a univariate and multivariate analysis approach that allowed us to show that adding other musical features to the model did not improve the correspondence to the neural data over and above spectral flux alone.

Despite their differences, we found strong correspondence between the dependent variables from the two types of analyses. Specifically, TRF correlations were strongly correlated with stimulation-tempo SRCoh, and this correlation was higher than for SRCoh at the first harmonic of the stimulation tempo for the amplitude envelope, derivative and beat onsets (*Figure 4—figure supplement 1*). Thus, despite being computed on a relatively broad range of frequencies, the TRF seems to be correlated with frequency-specific measures at the stimulation tempo. The strong correspondence between the two analysis approaches has implications for how users interpret their results. Although certainly not universally true, we have noticed a tendency for TRF users to interpret their results in terms of a convolution of an impulse response with a stimulus, whereas users of stimulus–response correlation or coherence tend to speak of entrainment of ongoing neural oscillations. The current results demonstrate that the two approaches produce similar results, even though the logic behind the techniques differs. Thus, whatever the underlying neural mechanism, using one or the other does not necessarily allow us privileged access to a specific mechanism.

### Conclusions

This study presented new insights into neural synchronization to natural music. We compared neural synchronization to different musical features and showed strongest neural responses to the spectral flux. This has important implications for research on neural synchronization to music, which has so far often quantified stimulus rhythm with what we would argue is a subpar acoustic feature – the amplitude envelope. Moreover, our findings demonstrate that neural synchronization is strongest for slower beat rates, and for predictable stimuli, namely familiar music with an easy-to-perceive beat.

## Materials and methods

### Participants

Thirty-seven participants completed the study (26 female, 11 male, mean age = 25.7 years, SD = 4.33 years, age range = 19–36 years). Target sample size for this was estimated using G*Power3, assuming 80% power for a significant medium-sized effect. We estimate a target sample size of 24 (+4) for within-participant condition comparisons and 32 (+4) for correlations, and defaulted to the larger value since this experiment was designed to investigate both types of effects. The values in parentheses were padding to allow for discarding ~15% of the recorded data. The datasets of three participants were discarded because of large artefacts in the EEG signal (see section *EEG data Preprocessing*), technical problems or for not following the experimental instructions. The behavioral and neural data of the remaining 34 participants were included in the analysis.

Prior to the EEG experiment, all participants filled out an online survey about their demographic and musical background using LimeSurvey (LimeSurvey GmbH, Hamburg, Germany, http://www.lime-survey.org). All participants self-identified as German speakers. Most participants self-reported normal

hearing (seven participants reported occasional ringing in one or both ears). Thirty-four participants were right- and three were left-handed. Musical expertise was assessed using the Goldsmith Music Sophistication Index (Gold-MSI; *Müllensiefen et al., 2014*). Participants received financial compensation for participating (Online: 2.50 €, EEG: 7€ per 30 min). All participants signed the informed consent before starting the experiment. The study was approved by the Ethics Council of the Max Planck Society Ethics Council in compliance with the Declaration of Helsinki (Application No: 2019_04).

## Stimuli

The stimulus set started from 39 instrumental versions of musical pieces from different genres, including techno, rock, blues, and hip-hop. The musical pieces were available in a *.wav* format on Qobuz Downloadstore (https://www.qobuz.com/de-de/shop). Each musical piece was segmented manually using Audacity (Version 2.3.3, Audacity Team, https://www.audacityteam.org) at musical phrase boundaries (e.g. between chorus and verse), leading to a pool of 93 musical segments with varying lengths between 14.4 and 38 s. We did not use the beat count from any publicly available beat-tracking softwares, because they did not track beats reliably across genres. Due to the first Covid-19 lockdown, we assessed the original tempo of each musical segment using an online method. Eight tappers, including the authors, listened to and tapped to each segment on their computer keyboard for a minimum of 17 taps; the tempo was recorded using an online BPM estimation tool (https://www.all8.com/tools/bpm.htm). In order to select stimuli with unambiguous strong beats that are easy to tap to, we excluded 21 segments due to high variability in tapped metrical levels (if more than 2 tappers tapped different from the others) or bad sound quality.

The remaining 72 segments were then tempo-manipulated using a custom-written MAX patch (Max 8.1.0, Cycling '74, San Francisco, CA, USA). Each segment was shifted to tempi between 1 and 4 Hz in steps of 0.25 Hz. All musical stimuli were generated using the MAX patch, even if the original tempo coincided with the stimulation tempo. Subsequently, the authors screened all of the tempo-shifted music and eliminated versions where the tempo manipulation led to acoustic distortions, made individual notes indistinguishable, or excessively repetitive. Overall, 703 music stimuli with durations of 8.3–56.6 s remained. All stimuli had a sampling rate of 44,100 Hz, were converted from stereo to mono, linearly ramped with 500 ms fade-in and fade-out and root-mean-square normalized using Matlab (R2018a; The MathWorks, Natick, MA, USA). A full overview of the stimulus segments, the original tempi and the modulated tempo range can be found in the Appendix (*Appendix 1—table 1*, *Figure 1—figure supplement 2*).

Each participant was assigned to one of four pseudo-randomly generated stimulus lists. Each list comprised 4–4.6 min of musical stimulation per tempo condition (*Kaneshiro et al., 2020*), resulting in 7–17 different musical segments per tempo and a total of 159–162 segments (trials) per participant. Each segment was repeated only once per tempo but was allowed to occur up to three times at different tempi within one experimental session (tempo difference between two presentations of the same segment was 0.5 Hz minimum). The presentation order of the musical segments was randomly generated for each participant prior to the experiment. The music stimuli were played at 50 dB sensation level (SL), based on individual hearing thresholds that were determined using the method of limits (*Leek, 2001*).

## Experimental design

After attaching the EEG electrodes and seating the participant in an acoustically and electrically shielded booth, the participant was asked to follow the instructions on the computer screen (BenQ Monitor XL2420Z, 144 Hz, 24", 1920 × 1080, Windows 7 Pro (64-bit)). The auditory and visual stimulus presentation was achieved using custom-written Matlab scripts using Psychtoolbox (PTB-3, *Brainard, 1997*) in Matlab (R2017a; The MathWorks, Natick, MA, USA). Upon publication the Source Code for stimulus presentation can be found on the projects OSF repository (*Weineck et al., 2022*).

The overall experimental flow for each participant can be found in *Figure 1A*. First, each participant conducted a self-paced spontaneous motor tempo task (SMT; *Fraisse, 1982*), which is a commonly used technique to assess individual's preferred tapping rate (*Rimoldi, 1951*, *Mcauley, 2010*). To obtain SMT, each participant tapped for thirty seconds (3 repetitions) at a comfortable rate with a finger on the table close to a contact microphone (Oyster S/P 1605, Schaller GmbH, Postbauer-Heng, Germany). Second, we estimated individual's hearing threshold using the method of limits. All sounds

in this study were delivered by a Fireface soundcard (RME Fireface UCX Audiointerface, Audio AG, Haimhausen, Germany) via on-ear headphones (Beyerdynamics DT-770 Pro, Beyerdynamic GmbH & Co. KG, Heilbronn, Germany). After a short three-trial training, the main task was performed. The music stimuli in the main task were grouped into eight blocks with approximately 20 trials per block and the possibility to take a break in between.

Each trial comprised two parts: attentive listening (music stimulation without movement) and tapping (music stimulation +finger tapping; *Figure 1A*). During attentive listening, one music stimulus was presented (8.3–56.6 s) while the participant looked at a fixation cross on the screen; the participant was instructed to mentally locate the beat without moving. Tapping began after a 1 s interval; the last 5.5 s of the previously listened musical segment were repeated, and participants were instructed to tap a finger to the beat of the musical segment (as indicated by the replacement of the fixation cross by a hand on the computer screen). Note that 5.5 s of tapping data is not sufficient to conduct standard analyses of sensorimotor synchronization; rather, our goal was to confirm that the participants tapped at the intended beat rate based on our tempo manipulation. After each trial, participants were asked to rate the segment based on *enjoyment/pleasure*, *familiarity* and *ease of tapping to the beat* with the computer mouse on a visual analogue scale ranging from –100 to +100. At the end of the experiment, the participant performed the SMT task again for three repetitions.

## EEG data acquisition

EEG data were acquired using BrainVision Recorder (v.1.21.0303, Brain Products GmbH, Gilching, Germany) and a Brain Products actiCap system with 32 active electrodes attached to an elastic cap based on the international 10–20 location system (actiCAP 64Ch Standard-2 Layout Ch1-32, Brain Products GmbH, Gilching, Germany). The signal was referenced to the FCz electrode and grounded at the AFz position. Electrode impedances were kept below 10 kOhm. The brain activity was acquired using a sampling rate of 1000 Hz via a BrainAmp DC amplifier (BrainAmp ExG, Brain Products GmbH, Gilching, Germany). To ensure correct timing between the recorded EEG data and the auditory stimulation, a TTL trigger pulse over a parallel port was sent at the onset and offset of each musical segment and the stimulus envelope was recorded to an additional channel using a StimTrak (StimTrak, Brain Products GmbH, Gilching, Germany).

## Data analysis
### Behavioral data

Tapping data were processed offline with a custom-written Matlab script. To extract the taps, the *.wav* files were imported and downsampled (from 44.1 kHz to 2205 Hz). The threshold for extracting the taps was adjusted for each trial manually (SMT and music tapping) and trials with irregular tap intervals were rejected. The SMT results were not analyzed as part of this study and will not be discussed further. For the music tapping, only trials with at least three taps (two intervals) were included for further analysis. Five participants were excluded from the music tapping analysis due to irregular and inconsistent taps within a trial (if >40% of the trials were excluded).

On each trial, participants were asked to rate the musical segments based on *enjoyment/pleasure*, *familiarity* and *ease to tap to the beat*. The rating scores were normalized to the maximum absolute rating per participant and per category. For the group analysis the mean and standard error of the mean (SEM) were calculated. For assessing the effects of each subjective dimension on neural synchronization, the 15 trials with the highest and lowest ratings (regardless of the tempo) per participant were further analyzed (see *EEG – Temporal Response Function*).

### Audio analysis

We assessed neural synchronization to four different musical features (*Figure 1B–C*). Note that the term 'musical feature' is used to describe time-varying features of music that operate on a similar time-scale as neural synchronization as opposed to the classical musical elements such as syncopation or harmony; (1) Amplitude envelope – gammatone filtered amplitude envelope in the main manuscript and absolute value of the full-band Hilbert envelope in the figure supplement; the gammatone filterbank consisted of 128 channels linearly spaced between 60 and 6000 Hz. (2) Half-wave rectified, first derivative of the amplitude envelope, which detects energy changes over time and is typically more sensitive to onsets (*Daube et al., 2019*; *Di Liberto et al., 2020*). (3) Binary-coded beat onsets

(0=no beat; 1=beat); a professionally trained percussionist tapped with a wooden drumstick on a MIDI drum pad to the beat of each musical segment at the original tempo (three trials per piece). After latency correction, the final beat times were taken as the average of the two takes with the smallest difference (*Harrison and Müllensiefen, 2018*). (4) Spectral novelty ('spectral flux') (*Müller, 2015*) was computed using a custom-written Python script (Python 3.6, Spyder 4.2.0) using the packages *numpy* and *librosa*. For computing the spectral flux of each sound, the spectrogram across frequencies of consecutive frames (frame length = 344 samples) was compared. The calculation of the spectral flux is based on the logarithmic amplitude spectrogram that results in a 1D vector (spectral information fluctuating over time). All stimulus features were z-scored and downsampled to 128 Hz for computing the stimulus-brain synchrony. To account for slightly different numbers of samples between stimulus features, they were cut to have matching sample sizes.

To validate that each musical feature contained acoustic cues to our tempo manipulation, we conducted a discrete Fourier transform using a Hamming window on each musical segment (resulting frequency resolution of 0.0025 Hz), averaged and z-scored the amplitude spectra per tempo and per musical feature (*Figure 1C*).

To assess how much information the different musical features share, a mutual information (MI) score was computed between each pair of musical features (*Figure 1D*). MI (in bits) is a time-sensitive measure that quantifies the reduction of uncertainty for one variable after observing a second variable (*Cover and Thomas, 2005*). MI was computed using *quickMI* from the Neuroscience Information Theory Toolbox with 4 bins, no delay, and a p-value cut-off of 0.001 (*Timme and Lapish, 2018*). For each stimulus feature, all trials were concatenated in the same order for each tempo condition and stimulation subgroup (Time x 13 Tempi x 4 Subgroups). MI values for pairs of musical features were compared to surrogate datasets in which one musical feature was time reversed (*Figure 1D*). To statistically assess the shared information between musical features, a three-way ANOVA test was performed (with first factor: data-surrogate comparison; second factor: tempo and third factor: stimulation subgroup).

## EEG data preprocessing

Unless stated otherwise, all EEG data were analyzed offline using custom-written Matlab code (R2019b; The MathWorks, Natick, MA, USA) combined with the Fieldtrip toolbox (*Oostenveld et al., 2011*). The continuous EEG data were bandpass filtered between 0.5 and 30 Hz (Butterworth filter), re-referenced to the average reference, downsampled to 500 Hz, and epoched between 1 s after stimulus onset (to remove onset responses to the start of the music stimulus) until the end of the initial musical segment presentation (attentive listening part of the trial). Single trials and channels containing large artefacts were removed based on an initial visual inspection. Missing channels were interpolated based on neighbouring channels with a maximum distance of 3 (*ft_prepare_neighbours*). Subsequently, Independent Component Analysis (ICA) was applied to remove artefacts and eye movements semi-automatically. After transforming the data back from component to electrode space, electrodes that exceeded 4 standard deviations of the mean squared data for at least 10% of the recording time were excluded. If bad electrodes were identified, pre-processing for that recording was repeated after removing the identified electrode (*Kaneshiro et al., 2020*). For the RCA analysis, if an electrode was identified for which 10% of the trial data exceeded a threshold of mean +2 standard deviations of the single-trial, single-electrode mean squared amplitude, the electrode data of the entire trial was replaced by NaNs. Next, noisy transients of the single-trial, single-electrode recordings were rejected. Therefore, data points were replaced by NaNs when the data points exceeded a threshold of two standard deviations of the single-trial, single-electrode mean squared amplitude. This procedure was repeated four times to ensure that all artefacts were removed (*Kaneshiro et al., 2020*). For the TRF analysis, which does not operate on NaNs, noisy transients were replaced by estimates using shape-preserving piecewise cubic spline interpolation or by the interpolation of neighbouring channels for single-trial bad electrodes.

Next, the data were restructured to match the requirements of the RCA or TRF (see sections *EEG – Temporal Response Function* and *EEG – Reliable Component Analysis*), downsampled to 128 Hz and z-scored. If necessary, the neural data were cut to match the exact sample duration of the stimulus feature per trial. For the RCA analysis approach, the trials in each tempo condition were concatenated resulting in a time-by-electrode matrix (Time x 32 Electrodes; with Time varying across tempo

condition). Subsequently the data of participants in the same subgroup were pooled together in a time-by-electrode-by-participant matrix (Time x 32 Electrodes x 9 or 10 Participants depending on the subgroup). In contrast to the RCA, for the TRF analysis, trials in the same stimulation condition were not concatenated in time, but grouped into cell arrays per participant according to the stimulus condition (Tempo x Trials x Electrodes x Time).

## EEG – reliable component analysis

To reduce data dimensionality and enhance the signal-to-noise ratio, we performed RCA (reliable components analysis, also correlated components analysis) (*Dmochowski et al., 2012*). RCA is designed to capture the maximum correlation between datasets of different participants by combining electrodes linearly into a vector space. One important feature of this technique is that it maximizes the correlation between electrodes across participants (which differentiates it from the similar canonical correlation analysis) (*Madsen et al., 2019*). Using the *rcaRun* Matlab function (*Dmochowski et al., 2012*; *Kaneshiro et al., 2020*), the time-by-electrode matrix was transformed to a time-by-component matrix with the maximum across-trial correlation in the first reliable component (RC1), followed by components with correlation values in descending order. For each RCA calculation, for each tempo condition and subgroup, the first three RCs were retained, together with forward-model projections for visualizing the scalp topographies. The next analysis steps in the time and frequency-domain were conducted on the maximally correlated RC1 component.

To examine the correlation between the neural signal and stimulus over time, the stimulus-response correlation (SRCorr) was calculated for every musical feature. This analysis procedure was adopted from *Kaneshiro et al., 2020*. In brief, every stimulus feature was concatenated in time with trials of the same tempo condition and subgroup to match the neural component-by-time matrix. The stimulus features were temporally filtered to account for the stimulus–brain time lag, and the stimulus features and neural time-courses were correlated. To create a temporal filter, every stimulus feature was transformed into a Toeplitz matrix, where every column repeats the stimulus-feature time course, shifted by one sample up to a maximum shift of 1 s, plus an additional intercept column. The Moore-Penrose pseudoinverse of the Toeplitz matrix and temporal filter was used to calculate the SRCorr. To report the SRCorr, the mean (± SEM) correlation coefficient across tempo conditions for every stimulus feature was calculated. For comparing tempo-specificity between musical features, a linear regression was fit to SRCorr values (and TRF correlations) as a function of tempo for every participant and for every musical feature (using *fitlm*). We compared the resulting slopes across musical features with a one-way ANOVA.

Stimulus-response coherence (SRCoh) is a measure that quantifies the consistency of phase and amplitude of two signals in a specific frequency band and ranges from 0 (no coherence) to 1 (perfect coherence) (*Srinivasan et al., 2007*). Here, the magnitude-squared coherence between different stimulus features and neural data was computed using the function *mscohere* with a Hamming window of 5 s and 50% overlap, resulting in a frequency range 0–64 Hz with a 0.125 Hz resolution. As strong coherence was found at the stimulation tempo and the first harmonic, the SRCoh values of each frequency vector were compared between musical features.

In order to control for any frequency-specific differences in the overall power of the neural data that could have led to artificially inflated observed neural synchronization at lower frequencies, the SRCorr and SRCoh values were z-scored based on a surrogate distribution (*Zuk et al., 2021*). Each surrogate distribution was generated by shifting the neural time course by a random amount relative to the musical feature time courses, keeping the time courses of the neural data and musical features intact. For each of 50 iterations, a surrogate distribution was created for each stimulation subgroup and tempo condition. The z-scoring was calculated by subtracting the mean and dividing by the standard deviation of the surrogate distribution.

## EEG – temporal response function

The TRF is a modeling technique, which computes a filter that optimally describes the relationship between the brain response and stimulus features (*Ding and Simon, 2012*; *Crosse et al., 2016*). Via linear convolution, the filter delineates how the stimulus features map onto the neural response (forward model), using ridge regression to avoid overfitting (range of lambda values: $10^{-6}$ - $10^{6}$). All computations of the TRF used the Matlab toolbox "The multivariate Temporal Response Function

(mTRF) Toolbox" (*Crosse et al., 2016*). The TRF was calculated in a leave-one-out cross-validation manner for all trials per stimulation tempo; this procedure was repeated for each musical feature separately, and additionally for all musical features together in a multivariate model (using *mTRF-crossval* and *mTRFtrain*) using time lags 0–400ms (*Di Liberto et al., 2020*). For the multivariate TRF approach, the stimulus features were combined by replacing the single time-lag vector by several time-lag vectors for every musical feature (Time x 4 musical features at different time lags). Using *mTRFpredict*, the neural time course of the left-out trial was predicted based on the time course of the corresponding musical feature of that trial. The quality of the predicted neural data was assessed by computing Pearson correlations between the predicted and actual EEG data separately for each electrode (TRF correlations). We averaged over the seven to eight electrodes with the highest TRF correlations that also corresponded to a canonical auditory topography. To quantify differences in the TRFs, the mean TRF correlation across stimulation tempo and/or musical feature was calculated per participant. The TRF weights across time lags were Fisher-z-scored (*Figure 3C–F*; *Crosse et al., 2016*). Analogous to the SRCorr and SRCoh, the TRF correlations were z-scored based on subtracting the mean and dividing the standard deviation of a surrogate distribution which was generated by shifting the neural data randomly relative to the musical features during the training and prediction of the TRF for 50 iterations per participant and stimulation tempo.

We tested the effects of more vs. less modulated music segments on the neural response by comparing TRF correlations within a stimulation tempo condition (*Figure 3—figure supplement 3*). Therefore, we took up to three trials per participant within the 2.25 Hz stimulation tempo condition where the original tempo ranged between (2.01–2.35 Hz) and compared them to up to three trials where the original tempo was slower (1.25–1.5 Hz). The same analysis was repeated in the 1.5 Hz stimulation tempo condition (original tempo ~1.25–1.6 Hz vs. originally faster music at ~2.1–2.5 Hz).

The assessment of TRF weights across time lags was accomplished by using a clustering approach for each musical feature and comparing significant data clusters to clusters from a random distribution (*Figure 3C–F*). To extract significant time windows in which the TRF weights were able to differentiate the different tempo conditions, a one-way ANOVA was performed at each time point. Clusters (consecutive time windows) were identified if the p-value was below a significance level of 0.01 and the size and F-statistic of those clusters were retained. Next, the clusters were compared to a surrogate dataset, which followed the same procedure, but had the labels of the tempo conditions randomly shuffled before entering it to the ANOVA. This step was repeated 1000 times (permutation testing). At the end, the significance of clusters was evaluated by subtracting the proportion of times the summed F-values of each clusters exceeded the summed F-values of the surrogate clusters from 1. A p-value below 0.05 was considered significant (*Figure 3G*). This approach yielded significant regions for the full-band (Hilbert) envelope and derivative (*Figure 3—figure supplement 1*). As these clusters did not show differences across amplitudes but rather in time, a latency analysis was conducted. Therefore, local minima around the grand average minimum or maximum within the significant time lag window were identified for every participant/tempo condition and the latencies retained. As there was no significant correlation between latencies and tempo conditions, the stimulation tempi were split upon visual inspection into two groups (1–2.5 Hz and 2.75–4 Hz). Subsequently, a piecewise linear regression was fitted to the data and the $R^2$ and p-values calculated (*Figure 3—figure supplement 1*).

In order to test whether spectral flux predicted the neural signal over and above the information it shared with the amplitude envelope, first derivative and beat onsets, we calculated TRFs for spectral flux after 'partialing out' their effects (*Figure 3—figure supplement 2*). This was achieved by first calculating TRF predictions based on a multivariate model comprising the amplitude envelope, derivate and beat onsets, and second, subtracting those predictions from the 'actual' EEG data and using the residual EEG data to compute a spectral flux model.

TRFs were evaluated based on participant ratings of enjoyment, familiarity, and ease to tap to the beat. Two TRFs were calculated per participant based on the 15 highest and 15 lowest ratings on each measure (ignoring tempo condition and subgroup), and the TRF correlations and time lags were compared between the two groups of trials (*Figure 5*). Significant differences between the groups were evaluated based on paired-sample t-tests.

The effect of musical sophistication was analyzed by computing the Pearson correlation coefficients between the maximum TRF correlation across tempi per participant and the general musical sophistication (Gold-MSI) per participant (*Figure 5—figure supplement 2*).

### EEG – comparison of TRF and RCA measures

The relationship between the TRF analysis approach and the SRCorr was calculated using a linear-mixed effects model (using *fitlme*). Participant and tempo were random (grouping) effects; SRCorr the fixed (predictor) effect and TRF correlations the response variable. To examine the underlying model assumption, the residuals of the linear-mixed effects model were plotted and checked for consistency. The best predictors of the random effects and the fixed-effects coefficients (beta) were computed for every musical feature and illustrated as violin plot (*Figure 4*).

## Statistical analysis

For each analysis, we assessed the overall difference between multiple subgroups using a one-way ANOVA. To test for significant differences across tempo conditions and musical features (TRF Correlation, SRCorr and SRCoh), repeated-measure ANOVAs were conducted coupled to Tukey's test and Greenhouse-Geiser correction was applied when the assumption of sphericity was violated (as calculated with the Mauchly's test). As effect size measures, we report partial $\eta^2$ for repeated-measures ANOVAs and $r_{equivalent}$ for paired sample t-test (*Rosenthal and Rubin, 2003*). Where applicable, the p-values were corrected using the False Discovery Rate (FDR).

## Acknowledgements

We thank the lab staff of the Max Planck Institute for Empirical Aesthetics for technical support during data acquisition and Lauren Fink for valuable input during data analysis and stimulus feature design.

## Additional information

### Funding

| Funder | Grant reference number | Author |
| --- | --- | --- |
| European Research Council | ERC-STG-804029 BRAINSYNC | Molly J Henry |
| Max-Planck-Gesellschaft | Max Planck Research Group Grant | Molly J Henry |

The funders had no role in study design, data collection and interpretation, or the decision to submit the work for publication.

### Author contributions

Kristin Weineck, Conceptualization, Software, Formal analysis, Investigation, Visualization, Methodology, Writing - original draft; Olivia Xin Wen, Software, Writing - review and editing; Molly J Henry, Conceptualization, Software, Formal analysis, Supervision, Funding acquisition, Methodology, Writing - original draft, Writing - review and editing

### Author ORCIDs

Kristin Weineck (iD) http://orcid.org/0000-0003-3204-860X
Olivia Xin Wen (iD) http://orcid.org/0000-0001-8845-1233
Molly J Henry (iD) http://orcid.org/0000-0002-2284-8884

### Ethics

All participants signed the informed consent before starting the experiment. The study was approved by the Ethics Council of the Max Planck Society Ethics Council in compliance with the Declaration of Helsinki (Application No: 2019_04).

### Decision letter and Author response

Decision letter https://doi.org/10.7554/eLife.75515.sa1

Author response https://doi.org/10.7554/eLife.75515.sa2

## Additional files

### Supplementary files
• Transparent reporting form

### Data availability

The source data and source code of all main results, the source code of the musical stimulus presentation and the raw EEG data are freely available on the OSF repository (https://doi.org/10.17605/OSF. IO/Y5XHS).

The following dataset was generated:

| Author(s) | Year | Dataset title | Dataset URL | Database and Identifier |
|---|---|---|---|---|
| Weineck K, Wen O, Henry MJ | 2022 | Neural synchronization is strongest to the spectral flux of slow music and depends on familiarity and beat salience | https://doi.org/10.17605/OSF.IO/Y5XHS | Open Science Framework, 10.17605/OSF.IO/Y5XHS |

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

# Appendix 1

**Appendix 1—table 1.** Overview over the music stimuli.

Parameters of stimulus creation for all 72 musical stimulus segments. The columns indicate 1. the stimulus number, 2. title of the musical piece, 3. the Artist of each musical piece, 4. the CD each piece was taken from (available at Qobuz Downloadstore), 5. timestamp of the music segment onset relative to the start of the recording [min.sec,ms], 6. duration of the music segment [sec] relative to the start of the music segment, 7. original tempo of excerpt [BPM; beats per minute] based on the taps of the authors and their colleagues and 8. frequency range [Hz] of the tempo-modulation (in 0.25 Hz steps) for each music piece.

| No. | Title | Artist | CD | Start [min] | Duration [sec] | Tempo [BPM/Hz] | | Range [Hz] |
|---|---|---|---|---|---|---|---|---|
| 1 | Abba Medley | Super Troopers | Instrumental Pop Hits | 7.30,71 | 22,99 | 136.09 / | 2.27 | 1.5–3.75 |
| 2 | Abba Medley | Super Troopers | Instrumental Pop Hits | 8.59,57 | 21,69 | 135.92 / | 2.27 | 1.75–4 |
| 3 | All is Alive | Francesco P. | Instrumental Hits, Vol.1 | 1.22,85 | 21,41 | 128.27 / | 2.14 | 1.5–4 |
| 4 | All is Alive | Francesco P. | Instrumental Hits, Vol.1 | 2.06,72 | 21,62 | 127.98 / | 2.13 | 1.5–4 |
| 5 | Apache | The Shadows | Rock Story "Instrumental Versions" | 0.39,59 | 14,78 | 135.22 / | 2.25 | 1.75–4 |
| 6 | Apache | The Shadows | Rock Story "Instrumental Versions" | 0.54,25 | 21,60 | 133.86 / | 2.23 | 1.75–4 |
| 7 | La Bikina | Rubén Fuentes Gasson | Bachata | 0.47,49 | 14,99 | 124.83 / | 2.08 | 1.75–4 |
| 8 | Bulldog | The Ventures | Rock Story "Instrumental Versions" | 0.06,15 | 18,13 | 151.33 / | 2.52 | 2–4 |
| 9 | Careless Whisper | Mads Haaber | Instrumental Pop Hits | 2.41,93 | 25,50 | 76.93 / | 1.28 | 1–2.75 |
| 10 | Cocaine | Corben Cassavette | Instrumental Pop Hits | 1.39,29 | 25,74 | 105.13 / | 1.75 | 1.5–4 |
| 11 | Dark Place | Beataddictz | Street Beatz, Vol.2 | 0.20,83 | 22,22 | 92.13 / | 1.54 | 1–3.75 |
| 12 | F.B. I. | The Shadows | Rock Story "Instrumental Versions" | 0.20,59 | 15,31 | 140.05 / | 2.33 | 1.25–4 |
| 13 | Five Trips | Tr3ntatr3 Giri | Instrumental Hits, Vol.1 | 1.48,79 | 16,48 | 123.24 / | 2.05 | 1.5–4 |
| 14 | Guybo | Eddie Cochran | Rock Story "Instrumental Versions" | 0.18,99 | 16,28 | 110.00 / | 1.83 | 1.5–3 |
| 15 | Gypsy Salsa, Cha Cha Beat | Corp Latino Dance Group | Hot Latin Dance | 1.57,06 | 24,26 | 100.04 / | 1.67 | 1.5–3 |
| 16 | Highway Riderz | Beataddictz | Street Beatz, Vol.2 | 0.19,61 | 30,30 | 97.13 / | 1.62 | 1–4 |
| 17 | In Go | Chuck Berry | Rock Story "Instrumental Versions" | 0.26,10 | 24,00 | 116.30 / | 1.94 | 1.5–4 |

*Appendix 1—table 1 Continued on next page*

*Appendix 1—table 1 Continued*

| No. | Title | Artist | CD | Start [min] | Duration [sec] | Tempo [BPM/Hz] | | Range [Hz] |
|---|---|---|---|---|---|---|---|---|
| 18 | Oh by Jingo! | Chet Atkins | Rock Story "Instrumental Versions" | 0.07,47 | 23,59 | 120.55 / | 2.01 | 1–3.5 |
| 19 | Keep It 1,000 | Beataddictz | Street Beatz, Vol.2 | 1.13,87 | 25,10 | 78.06 / | 1.30 | 1–3 |
| 20 | The Last Day | Beataddictz | Street Beatz, Vol.2 | 1.27,40 | 22,52 | 88.13 / | 1.47 | 1–2.5 |
| 21 | For the Last Time | Beataddictz | Street Beatz, Vol.2 | 1.16,88 | 26,21 | 74.86 / | 1.25 | 1–3.25 |
| 22 | Lights Out | Beataddictz | Street Beatz, Vol.2 | 0.32,12 | 21,60 | 89.12 / | 1.49 | 1.25–3.5 |
| 23 | I like | Francesco P. | Instrumental Hits, Vol.1 | 1.34,19 | 27,95 | 112.13 / | 1.87 | 1.25–3.75 |
| 24 | Dark Line | Alex Cundari | Instrumental Hits, Vol.1 | 0.37,81 | 19,75 | 106.08 / | 1.77 | 1.25–3 |
| 25 | Live Forever | The Wonderwalls | Instrumental Pop Hits | 1.16,28 | 22,61 | 90.11 / | 1.50 | 1.5–3.75 |
| 26 | Lucy in the Sky with Diamonds | Ricardo Caliente | Instrumental Pop Hits | 2.43,55 | 23,93 | 81.11 / | 1.35 | 1–3.5 |
| 27 | Monalisa | Ken Laszlo | Instrumental Hits, Vol.1 | 1.13,37 | 29,79 | 129.97 / | 2.17 | 1.5–4 |
| 28 | Monalisa | Ken Laszlo | Instrumental Hits, Vol.1 | 2.13,01 | 28,80 | 130.02 / | 2.17 | 1.5–4 |
| 29 | Can't Fight the Moonlight | Jon Carran | Instrumental Pop Hits | 1.10,02 | 18,56 | 97.95 / | 1.63 | 1.5–3.75 |
| 30 | Muy Tranquilo | Gramatik | Muy Tranquilo | 2.05,1 | 26,59 | 90.04 / | 1.50 | 1–3.25 |
| 31 | No Mercy | Beataddictz | Street Beatz, Vol.2 | 0.12,15 | 26,41 | 76.11 / | 1.27 | 1–3.5 |
| 32 | I'm A Pusha | Beataddictz | Street Beatz, Vol.2 | 0.11,07 | 22,60 | 85.15 / | 1.42 | 1–3.25 |
| 33 | Rockin' the Blues Away | Tiny Grimes Quintet | Rock Story "Instrumental Versions" | 0.05,61 | 20,84 | 141.05 / | 2.35 | 1.5–4 |
| 34 | The Rocking Guitar | Ini Kamoze | Rock Story "Instrumental Versions" | 0.17,39 | 16,21 | 118.66 / | 1.98 | 1.5–3.25 |
| 35 | Country Rodeo Song | Marco Rinaldo | Country Instrumental Mix | 1.46,35 | 27,70 | 112.94 / | 1.88 | 1.5–3.75 |
| 36 | I Shot the Sheriff | Corben Cassavette | Instrumental Pop Hits | 0.19,52 | 25,54 | 94.12 / | 1.57 | 1.25–4 |
| 37 | Sing Sing Sing | Benny Goodman | Sing Sing Sing | 0.18,23 | 36,01 | 108.68 / | 1.81 | 1.5–3 |
| 38 | Si Una Vez | Pete Astudillo | Bachata | 0.46,54 | 16,95 | 124.54 / | 2.08 | 1.5–3 |
| 39 | I'm Still Standing | Ricardo Caliente | Instrumental Pop Hits | 0.39,14 | 21,72 | 86.04 / | 1.43 | 1.25–2.75 |

*Appendix 1—table 1 Continued on next page*

*Appendix 1—table 1 Continued*

| No. | Title | Artist | CD | Start [min] | Duration [sec] | Tempo [BPM/Hz] | | Range [Hz] |
|---|---|---|---|---|---|---|---|---|
| 40 | Streets On Fire | Beataddictz | Street Beatz, Vol.2 | 0.23,33 | 24,99 | 81.16 / | 1.35 | 1–3.75 |
| 41 | Tequila | The Champs | Rock Story "Instrumental Versions" | 1.02,85 | 21,45 | 89.40 / | 1.49 | 1–3 |
| 42 | Vegas Dream | Vegas Project | Instrumental Hits, Vol.1 | 1.13,73 | 22,73 | 128.08 / | 2.13 | 1.5–3.25 |
| 43 | I Can't Wait | Alex Cundari | Instrumental Hits, Vol.1 | 0.23,73 | 24,08 | 83.59 / | 1.39 | 1–2.5 |
| 44 | Who Dat | Beataddictz | Street Beatz, Vol.2 | 1.15,58 | 24,50 | 87.10 / | 1.45 | 1–3.25 |
| 45 | Abba Medley | Super Troopers | Instrumental Pop Hits | 0.28,38 | 27,05 | 136.09 / | 2.27 | 1.25–4 |
| 46 | Abba Medley | Super Troopers | Instrumental Pop Hits | 5.09,89 | 21,71 | 136.09 / | 2.27 | 1–3 |
| 47 | Abba Medley | Super Troopers | Instrumental Pop Hits | 6.03,59 | 20,64 | 136.09 / | 2.27 | 1.5–4 |
| 48 | La Bikina | Rubén Fuentes Gasson | Bachata | 1.18,31 | 46,00 | 124.83 / | 2.08 | 1.75–4 |
| 49 | Bulldog | The Ventures | Rock Story "Instrumental Versions" | 0.44,98 | 19,20 | 151.33 / | 2.52 | 2–4 |
| 50 | Bulldog | The Ventures | Rock Story "Instrumental Versions" | 1.21,25 | 38,85 | 151.33 / | 2.52 | 2–4 |
| 51 | Careless Whisper | Mads Haaber | Instrumental Pop Hits | 3.09,44 | 24,41 | 76.93 / | 1.28 | 1–2.75 |
| 52 | Dark Place | Beataddictz | Street Beatz, Vol.2 | 1.45,73 | 35,10 | 92.13 / | 1.54 | 1–3.75 |
| 53 | F.B.I. | The Shadows | Rock Story "Instrumental Versions" | 0.37,30 | 20,91 | 140.05 / | 2.33 | 1.5–4 |
| 54 | Guybo | Eddie Cochran | Rock Story "Instrumental Versions" | 0.36,31 | 17,53 | 110.00 / | 1.83 | 1.5–3 |
| 55 | Highway Riderz | Beataddictz | Street Beatz, Vol.2 | 0.59,19 | 20,40 | 97.13 / | 1.62 | 1–4 |
| 56 | In Go | Chuck Berry | Rock Story "Instrumental Versions" | 0.48,42 | 26,80 | 116.30 / | 1.94 | 1.5–4 |
| 57 | Oh by Jingo! | Chet Atkins | Rock Story "Instrumental Versions" | 0.40,89 | 23,00 | 120.55 / | 2.01 | 1.25–3.5 |
| 58 | Live Forever | The Wonderwalls | Instrumental Pop Hits | 1.41,02 | 34,00 | 90.11 / | 1.50 | 1.25–3.5 |
| 59 | Live Forever | The Wonderwalls | Instrumental Pop Hits | 3.35,72 | 25,80 | 90.11 / | 1.50 | 1.25–3.5 |
| 60 | Lucy in the Sky with Diamonds | Ricardo Caliente | Instrumental Pop Hits | 3.06,40 | 23,00 | 81.11 / | 1.35 | 1–3.5 |

*Appendix 1—table 1 Continued on next page*

*Appendix 1—table 1 Continued*

| No. | Title | Artist | CD | Start [min] | Duration [sec] | Tempo [BPM/Hz] | | Range [Hz] |
|-----|-------|--------|-----|-----------|--------------|---------------|---|-----------|
| 61 | Can't Fight the Moonlight | Jon Carran | Instrumental Pop Hits | 1.42,26 | 21,82 | 97.95 / | 1.63 | 1.5–3.75 |
| 62 | No Mercy | Beataddictz | Street Beatz, Vol.2 | 0.50,16 | 26,74 | 76.11 / | 1.27 | 1–3.5 |
| 63 | No Mercy | Beataddictz | Street Beatz, Vol.2 | 2.33,53 | 26,41 | 76.11 / | 1.27 | 1–3.5 |
| 64 | Rockin' the Blues Away | Tiny Grimes Quintet | Rock Story "Instrumental Versions" | 0.47,65 | 25,50 | 141.05 / | 2.35 | 1.5–4 |
| 65 | Rockin' the Blues Away | Tiny Grimes Quintet | Rock Story "Instrumental Versions" | 1.12,78 | 36,07 | 141.05 / | 2.35 | 1.5–4 |
| 66 | The Rocking Guitar | Ini Kamoze | Rock Story "Instrumental Versions" | 0.33,58 | 15,08 | 118.66 / | 1.98 | 1.5–3.25 |
| 67 | Country Rodeo Song | Marco Rinaldo | Country Instrumental Mix | 2.13,99 | 24,03 | 112.94 / | 1.88 | 1.5–3.75 |
| 68 | Sing Sing Sing | Benny Goodman | Sing Sing Sing | 1.08,65 | 18,87 | 108.68 / | 1.81 | 1.5–3 |
| 69 | Sing Sing Sing | Benny Goodman | Sing Sing Sing | 2.46,03 | 36,29 | 108.68 / | 1.81 | 1.5–3 |
| 70 | Si Una Vez | Pete Astudillo | Bachata | 1.03,70 | 29,27 | 124.54 / | 2.08 | 1.5–3 |
| 71 | Streets on fire | Beataddictz | Street Beatz, Vol.2 | 2.00,63 | 38,00 | 81.16 / | 1.35 | 1–3.75 |
| 72 | I Can't Wait | Alex Cundari | Instrumental Hits, Vol.1 | 1.10,71 | 34,48 | 83.59 / | 1.39 | 1–2.5 |

