## [Editor Report]

This study investigated the neural tracking of music using novel methodology. The core finding was stronger neuronal entrainment to "spectral flux" rather than other more commonly tested features such as amplitude envelope. The study is methodologically sophisticated and provides novel insight on the neuronal mechanisms of music perception.

---

## [Decision Letter]

**Decision letter after peer review:**

Thank you for submitting your article "Neural entrainment is strongest to the spectral flux of slow music and depends on familiarity and beat salience" for consideration by *eLife*. Your article has been reviewed by 3 peer reviewers, and the evaluation has been overseen by a Reviewing Editor and Barbara Shinn-Cunningham as the Senior Editor. The following individuals involved in review of your submission have agreed to reveal their identity: Benedikt Zoefel (Reviewer #1) and Nate Zuk (Reviewer #3).

Essential revisions:

In general the reviewers were positive however more work needs to be done to validate that the results are not a consequence of the analyses or the specific choice of music before tempo manipulation as pointed out by the reviewers. Also, it would be important to better clarify the concepts of neural entertainment, synchronization and neural tracking and their meaning in this specific context.

*Reviewer #1 (Recommendations for the authors):*

I believe the authors could emphasize their most important results more when they appear in the Results section. An example is the beginning of page 11 where one important sentence ("SRCoh was highest…") is hidden in the text. This result is interesting – highlighting it would also make main results easier to extract.

RCA: "These approaches have been criticized because of their potential susceptibility to autocorrelation". As the authors mention a possible involvement of neural oscillations (in their introduction), it could be useful to point out that a removal of autocorrelation (to calculate TRF) might actually remove oscillations as well.

Due to differences in x and y axes, I was initially confused by Figure 1c, wondering why stimulation tempo (Hz) does not correspond to FFT frequency (Hz). Maybe the authors could include lines to show where stimulation tempo = FFT frequency and the first harmonic? This would make the relevant information much easier to extract.

I leave it to the authors, but an additional point that might be worth discussing is the fact that humans seem to be most sensitive to amplitude modulations at higher frequencies (around 4 Hz) than those that seem to play an important role in the current study (1-2 Hz). This is for example summarized in a review by Edwards and Chang (2013, Hear Res). Other relevant work is that by Teng, Poeppel and colleagues showing theta activity to most reliably follow acoustic rhythms. Could the authors discuss whether this means that music is special or other reasons for relatively low preferred rates in their work?

*Reviewer #2 (Recommendations for the authors):*

The paper was very nice but sometimes hard to read because I am not so confident with the difference between engagement, entertainment, synchronization and tracking. In the literature, those terms are sometimes used interchangeably, and sometimes instead, they are used with a precise meaning. I suggest the authors think about rephrasing parts of the paper and clarifying those terms from the beginning to broaden the range of readers that can enjoy the paper in depth. I found very few typos, and the visualizations were nice. Sometimes I think that the plots are too small and hard to read, and the x/y axes proportion must be chosen carefully.

*Reviewer #3 (Recommendations for the authors):*

p. 4, line 72: "…and we are not aware…tempo-modulated". I think Kaneshiro et al., 2020 fits this description. They did not use a controlled spacing of tempos like was done in this study, but they used naturalistic polyphonic music with a variety of tempos. Rajendran et al., 2020 also fits this description, although they did extracellular recordings and not EEG.

p. 6, line 106-107: "…spectral flux is sensitive…changes in amplitude." Can you provide a citation where rhythmic information is provided by changes in pitch but not in amplitude, perhaps including a behavioral measure of rhythm salience?

Figure 1C: I think the reason the z-scored amplitude decreases with decreasing stimulation tempo is because z-scoring reduces the magnitude of the components as more non-zero spectral components are introduced in the range of interest. This could be confusing to readers given that EEG may track these tempos better. The authors could alternatively z-score the original signals (before computing the spectra), which might result in more consistent amplitudes across stimulation tempo.

p. 20, lines 368-370: "In this study…tapped frequency." I think this should go earlier in the results, around where Figure 1 is presented, in order to clarify the difference between "stimulation tempo" (which is referred to earlier) and "tapped beat rate".

p. 25-26, lines 481-510: This discussion aims to highlight the importance of considering other stimulus features besides the envelope (like spectral flux) when comparing the neural tracking of speech and music, partly via criticizing Zuk et al. (2021) for focusing on the envelope. However, the current study does not fully back up some of the author's critiques, mainly because speech stimuli were not included in this study. We don't know how much spectral flux improves EEG prediction of speech over envelope-based measures, although it likely will (see the use of spectrotemporal changes and acoustic edges in Daube et al., 2019). But we can instead compare the prediction accuracies to Di Liberto et al., 2015, Current Biology.

Envelope-based forward modeling of speech produces the worst average prediction accuracies around 0.05, and the accuracies increase as features such as the spectrogram and phonetic features are added (Figure 2 of that paper). These prediction accuracies are above the prediction accuracies in the current study (0.04 on average with spectral flux, and 0.06 for the slowest tempos, see Figure 3 of this study), supporting the possibility that speech is generally tracked better than music. I agree it is possible that other musical features may predict EEG better and could produce comparable prediction accuracies to speech. But for the discussion here, the authors should instead focus on how similarities and differences in relevant features for speech and music could affect interpretations for studies comparing neural tracking, which would nicely follow the first paragraph of this section. More specifically, the authors should clarify what "characterize stimuli as fairly as possible" (line 509) means and suggest alternative ways of comparing speech and music. If they choose to criticize studies using the envelope alone (which might not be necessary to make their point), I recommend also acknowledging that their study does not directly address questions about speech and music comparisons and that more work needs to be done to understand the differences between speech and music tracking.

p. 25, lines 497-498: "…we found…envelope." It is not clear if spectral flux causes stronger entrainment given the data. Rephrase to: "…we found that spectral flux was tracked better than the amplitude envelope."

p. 26, lines 520-521: "…and ease of beat tapping…stimulation tempi." The statistical analysis of the data in Figure 3 showed that ease of beat tapping did not significantly vary with stimulation tempo (lines 327-329), which contradicts this statement.

p. 28, lines 532-535: "One is that our study…entrainment." I don't understand why this would matter, because the analysis was done after selecting "musical" and "non-musical" subjects. Perhaps cut this?

p. 29, line 594: "correlate" should be "correlated".

p. 32, line 647-648: "Each segment…0.25 Hz." The details for the range of tempi are in Supplementary Table 1, but it would be useful here to state the maximum amount of tempo shift relative to the original, both for increasing and decreasing the tempo. Additionally, I mentioned in the public review that the original tempos seem to span the range with the best neural tracking. You should also state the original range of tempos here.

p. 34, lines 703-704: "This signal…AFz position." Why did you choose to reference this way? Is there a related citation?

p. 37, line 773: "To statistically asses" should be "assess".

p. 40, section EEG – Reliable Components Analysis, Temporal Response Function: In the public review, I suggested normalizing the measures of correlation, coherence, and prediction accuracy by null distributions. When you do so, make sure the shuffling of trials is done using trials with the same tempi. Otherwise, there will be a mismatch in the spectral content of the music due to differences in tempi, which defeats the point of generating the null distributions.

p. 40, line 855: "…a system identification technique…" This isn't quite true, change to "…a modeling technique…".

p. 41, lines 873-884, also Figure 3C: From my understanding, TRFs were fit to each stimulus feature separately. I am ok with this for comparing prediction accuracies because the models appear to have the same dimensionality (T x 1, where T is the number of lags). However, this is potentially an issue for interpreting TRF weights because the stimulus features in the separate models are correlated, so peaks and troughs found in the TRF for one model might be the effect of one of the other features. One possibility is to look at the TRF weights of the full model instead, but ridge unfortunately optimizes the model by making the weights correlated (an effect that produces smooth TRFs), so I don't think it alleviates the issue either. The best approach would be to iteratively partial out the contribution of each stimulus feature. Start with the envelope model, compute the EEG prediction with that model, and then subtract the prediction from the EEG data. Then fit the derivative model and then the beats model, removing each after they are fit. This way, the TRF weights found for the spectral flux model reflect the temporal response after removing the effects of the other stimulus features.

p. 42-43, lines 903-933: In the public review, I mentioned that the TRF-SVM modeling needed to be clearer. Specifically:

– How are the TRFs fit for each group? If you are using a leave-one-trial-out procedure like before, the resulting TRFs for each trial are the models fit to all trials except the testing trial. As a result, they are going to be highly similar to each other, which could explain why the SVM performs so well. Here, I recommend instead fitting the TRF to each trial separately (no cross-validation) using the same ridge parameter that you found for the model trained earlier (in the section EEG – Temporal Response Function). That way the TRFs will represent the stimulus-response mappings for each individual trial.

– Clarify which training step was referred to for calculating the surrogate data (line 920). I recommend shuffling prior to the TRF modeling step, although if you fit the TRF to each trial individually then the result of shuffling before vs after will be the same.

– Lines 913-916: There are 6 trials at tapped rate, and 6 trials at 2x tapped rate, which should result in n=12 not n=13.

Figure 5 – supplement 1: The distributions on the left don't look like they come from the points on the right in each plot. Can you check if there was a mistake with the points plotted or the y-axes?

---

## [Author Response]

Essential revisions:In general the reviewers were positive however more work needs to be done to validate that the results are not a consequence of the analyses or the specific choice of music before tempo manipulation as pointed out by the reviewers. Also, it would be important to better clarify the concepts of neural entertainment, synchronization and neural tracking and their meaning in this specific context.Reviewer #1 (Recommendations for the authors):I believe the authors could emphasize their most important results more when they appear in the Results section. An example is the beginning of page 11 where one important sentence ("SRCoh was highest…") is hidden in the text. This result is interesting – highlighting it would also make main results easier to extract.

We tried to highlight our most important results by dividing the results into smaller paragraphs. This way, we hope that the main results will be easier to read.

RCA: "These approaches have been criticized because of their potential susceptibility to autocorrelation". As the authors mention a possible involvement of neural oscillations (in their introduction), it could be useful to point out that a removal of autocorrelation (to calculate TRF) might actually remove oscillations as well.

This is true. We added this concern to the discussion of the manuscript.

p. 28 l. 556-560: “However, the RCA-based approaches (Kaneshiro et al., 2020) have been criticized because of their potential susceptibility to autocorrelation, which is argued to be minimized in the TRF approach (Zuk et al., 2021), which uses ridge regression to dampen fast oscillatory components (Crosse et al., 2021). However, by minimizing the effects of auto-correlation one concern could be that this could remove neural oscillations as well.”

Due to differences in x and y axes, I was initially confused by Figure 1c, wondering why stimulation tempo (Hz) does not correspond to FFT frequency (Hz). Maybe the authors could include lines to show where stimulation tempo = FFT frequency and the first harmonic? This would make the relevant information much easier to extract.

We added lines to Figure 1C to highlight when the first harmonic or the stimulation tempo equal the FFT Frequency.

I leave it to the authors, but an additional point that might be worth discussing is the fact that humans seem to be most sensitive to amplitude modulations at higher frequencies (around 4 Hz) than those that seem to play an important role in the current study (1-2 Hz). This is for example summarized in a review by Edwards and Chang (2013, Hear Res). Other relevant work is that by Teng, Poeppel and colleagues showing theta activity to most reliably follow acoustic rhythms. Could the authors discuss whether this means that music is special or other reasons for relatively low preferred rates in their work?

Thanks for this. It is true that Edwards and Chang, 2013 (among others) identify highest sensitivity to amplitude modulation around 4 Hz. This is faster than the rates at which we saw strongest neural synchronization to the amplitude envelope of music. It is possible then, that the rates at which we examined neural synchronization were “suboptimal” with respect to the system’s sensitivity to amplitude modulation, which may have handicapped amplitude envelope as a feature to describe music. However, Edwards and Chang also identify highest sensitivity to frequency modulation around the same rate, that is, 2–5 Hz. Again, this is faster than the rates at which we saw strongest neural synchronization to spectral flux. So we would argue that the difference between our amplitude and spectral features was not due to differences in the pre-existing sensitivities of the auditory system to these types of modulations. We actually have some data from another study (unpublished) showing that behavioral preferences for different rates are category-specific, meaning that you can tell the difference between amplitude- / frequency-modulated sounds, speech, and music based on the rates that listeners prefer to hear those sounds presented at. Although these are not neural data, they suggest to us that sensitivity to the modulations in technical sounds (AM, FM) might not be sufficient to predict sensitivity to fluctuations in categories of natural sounds, and music in particular. However, we would not necessarily propose that music is special in this way. Although we find this to be an extremely interesting topic, any discussion we would add would be fairly wild speculation – therefore, we hope to have your support by not adding this interesting point to our manuscript and lengthening the discussion further.

Nonetheless, we do include a discussion about why these low rates are important in natural music, which we reproduce here for your convenience.

p. 23, l. 409-426: “Strongest neural synchronization was found in response to stimulation tempi between 1 and 2 Hz in terms of SRCorr (Figure 2B), TRF correlations (Figure 3A), and TRF weights (Figure 3C-F). Moreover, we observed a behavioral preference to tap to the beat in this frequency range, as the group preference for music tapping was at 1.55 Hz (Figure 5 —figure supplement 3). Previous studies have shown a preference to listen to music with beat rates around 2 Hz (Bauer et al., 2015), which is moreover the modal beat rate in Western pop music (Moelants, 2002) and the rate at which the modulation spectrum of natural music peaks (Ding et al., 2017). Even in nonmusical contexts, spontaneous adult human locomotion is characterized by strong energy around 2 Hz (MacDougall and Moore, 2005). Moreover, when asked to rhythmically move their bodies at a comfortable rate, adults will spontaneously move at rates around 2 Hz (McAuley et al., 2006) regardless whether they use their hands or feet (Rose et al., 2020). Thus, there is a tight link between preferred rates of human body movement and preferred rates for the music we make and listen to that was moreover reflected in our neural data. This is perhaps not surprising, as musical rhythm perception activates motor areas of the brain, such as the basal ganglia and supplementary motor area (Grahn and Brett, 2007), and is further associated with increased auditory–motor functional connectivity (Chen et al., 2008). In turn, involving the motor system in rhythm perception tasks improves temporal acuity (Morillon et al., 2014), but only for beat rates in the 1–2 Hz range (Zalta et al., 2020).”

Reviewer #2 (Recommendations for the authors):The paper was very nice but sometimes hard to read because I am not so confident with the difference between engagement, entertainment, synchronization and tracking. In the literature, those terms are sometimes used interchangeably, and sometimes instead, they are used with a precise meaning. I suggest the authors think about rephrasing parts of the paper and clarifying those terms from the beginning to broaden the range of readers that can enjoy the paper in depth. I found very few typos, and the visualizations were nice. Sometimes I think that the plots are too small and hard to read, and the x/y axes proportion must be chosen carefully.

Thank you for your helpful comments and your positive review. We rephrased parts of the paper and moved explanatory sections from the discussion to the introduction. We decided to use the term “neural synchronization” throughout, but do discuss how our study relates to the concept of neural entrainment, and how the analysis frameworks we tested relate to different theoretical backgrounds. We also rearranged some Figures (especially the Figures 2 and 3) that appeared to be quite small. We hope that this way the Figures are better readable and easier to understand.

Reviewer #3 (Recommendations for the authors):p. 4, line 72: "…and we are not aware…tempo-modulated". I think Kaneshiro et al., 2020 fits this description. They did not use a controlled spacing of tempos like was done in this study, but they used naturalistic polyphonic music with a variety of tempos. Rajendran et al., 2020 also fits this description, although they did extracellular recordings and not EEG.

Kaneshiro et al., 2020 used four natural (unmanipulated) songs, which had tempi of 156 BPM, 94 BPM, 90 BPM and 86 BPM. Although it is true that this covers a range of tempi, the experimental manipulation was not to parametrically vary tempo as we did here. In contrast, Rajendran et al., 2020 used a larger musical stimulus set with tempi ranging from 0.7 to 3.7 Hz and investigated the neural response to those musical stimuli in rats.

Despite these differences, we can see why those studies could fit this description in our Introduction, and we have changed the sentence.

p. 3 l. 54-57: “Despite the perceptual and motor evidence, studies looking at tempo-dependence of neural synchronization are scarce (Doelling and Poeppel, 2015, Nicolaou et al., 2017) and we are not aware of any human EEG study using naturalistic polyphonic musical stimuli that were manipulated in the tempo domain.“

p. 6, line 106-107: "…spectral flux is sensitive…changes in amplitude." Can you provide a citation where rhythmic information is provided by changes in pitch but not in amplitude, perhaps including a behavioral measure of rhythm salience?

A number of past studies (such as Jones, 1987; Jones and Pfordresher, 1997; Ellis and Jones, 2009) describe the “joint accent structure” of music, where the position of perceived accents is determined by changes in pitch (deviations) in relation to the pre-existing melody (melodic accents) in addition to timing changes in relation of the temporal flow of an auditory stimulus (temporal accents, Jones and Pfordresher, 1997).

For a more concrete example, you could imagine classical music played in a glissando style by a violin or cello. There may never be a clear sound on-/offset, and the perceived rhythm can be based entirely on spectral changes of the music piece.

Finally, related to the behavioral consequences, a study by Burger et al., 2013, demonstrated a positive correlation between the spectral flux at lower frequencies (50-100 Hz) in music with perceived beat strength and urge to move.

We briefly mentioned parts of this in the discussion, but partially moved it up to the introduction and added citations:

p. 5 l. 95-100: “One potential advantage of spectral flux over the envelope or its derivative is that spectral flux is sensitive to rhythmic information that is communicated by changes in pitch even when they are not accompanied by changes in amplitude. Critically, temporal and spectral information jointly influence the perceived accent structure in music, which provides information about beat locations (Pfordresher, 2003, Ellis and Jones, 2009, Jones, 1993).”

p. 25 l. 476-477: “Previous work on joint accent structure indicates that spectral information is an important contributor to beat perception (Ellis and Jones, 2009, Pfordresher, 2003).”

Figure 1C: I think the reason the z-scored amplitude decreases with decreasing stimulation tempo is because z-scoring reduces the magnitude of the components as more non-zero spectral components are introduced in the range of interest. This could be confusing to readers given that EEG may track these tempos better. The authors could alternatively z-score the original signals (before computing the spectra), which might result in more consistent amplitudes across stimulation tempo.

We z-scored the original signal before computing the FFT. For plotting, we z-scored the FFT again. However, we compared the z-scored FFTs to the FFTs that were computed without subsequent z-scoring and similar trends were found (please see plots in Author response image 1). Therefore, we decided to keep the z-scored FFT (as shown in the original version of the manuscript).

**Author response image 1. sa2fig1:** 

p. 20, lines 368-370: "In this study…tapped frequency." I think this should go earlier in the results, around where Figure 1 is presented, in order to clarify the difference between "stimulation tempo" (which is referred to earlier) and "tapped beat rate".

Thank you. As we removed most of the tapping-related analysis from the manuscript, we also deleted this sentence.

p. 25-26, lines 481-510: This discussion aims to highlight the importance of considering other stimulus features besides the envelope (like spectral flux) when comparing the neural tracking of speech and music, partly via criticizing Zuk et al. (2021) for focusing on the envelope. However, the current study does not fully back up some of the author's critiques, mainly because speech stimuli were not included in this study. We don't know how much spectral flux improves EEG prediction of speech over envelope-based measures, although it likely will (see the use of spectrotemporal changes and acoustic edges in Daube et al., 2019). But we can instead compare the prediction accuracies to Di Liberto et al., 2015, Current Biology.Envelope-based forward modeling of speech produces the worst average prediction accuracies around 0.05, and the accuracies increase as features such as the spectrogram and phonetic features are added (Figure 2 of that paper). These prediction accuracies are above the prediction accuracies in the current study (0.04 on average with spectral flux, and 0.06 for the slowest tempos, see Figure 3 of this study), supporting the possibility that speech is generally tracked better than music. I agree it is possible that other musical features may predict EEG better and could produce comparable prediction accuracies to speech. But for the discussion here, the authors should instead focus on how similarities and differences in relevant features for speech and music could affect interpretations for studies comparing neural tracking, which would nicely follow the first paragraph of this section. More specifically, the authors should clarify what "characterize stimuli as fairly as possible" (line 509) means and suggest alternative ways of comparing speech and music. If they choose to criticize studies using the envelope alone (which might not be necessary to make their point), I recommend also acknowledging that their study does not directly address questions about speech and music comparisons and that more work needs to be done to understand the differences between speech and music tracking.

Thank you for this important comment and sorry for the potential misunderstanding that originated from this part of the discussion. In this section of the discussion, we did not mean to write negatively about studies that solely focus on the stimulus amplitude envelope. And we certainly did not mean to imply that our paper has anything definitive to say about direct comparisons of neural synchronization to music and speech. Every paper follows a different agenda and focuses on different aspects of the research in the field. In the current study, we wanted to test the effects of different acoustic features on neural synchronization (to music only). The point we were trying to make is not that the amplitude envelope is a “bad” acoustic feature, but rather that one could also consider using different acoustic features (going beyond the often-used amplitude envelope) to arrive at a more nuanced understanding of neural synchronization to music. As you say, this approach has been taken in speech work in the past and has improved forward-model predictions of neural data. Here, we are attemping a similar approach in the musical domain, where our choice of musical features was grounded in some intuition or understanding of the kinds of acoustic fluctuations that might give rise to a sense of temporal regularity or pulse.

Our aim with this discussion was not to create a conflict or a debate in the literature, but rather just to provide a little food for thought for future work. For that reason, we have substantially cut down this part of the discussion, and no longer speak of “characterizing stimuli as fairly as possible”. We reproduce this revised bit of the discussion here for your convenience:

p. 26-27 l. 498-511: “For example, a recent study found that neuronal activity synchronizes less strongly to music than to speech (Zuk et al., 2021); notably this paper focused on the amplitude envelope to characterize the rhythms of both stimulus types. However, our results show that neural synchronization is especially strong to the spectral content of music, and that spectral flux may be a better measure for capturing musical dynamics than the amplitude envelope (Müller, 2015). Imagine listening to a melody played in a glissando fashion on a violin. There might never be a clear onset that would be represented by the amplitude envelope – all of the rhythmic structure is communicated by spectral changes. Indeed, many automated tools for extracting the beat in music used in the musical information retrieval (MIR) literature rely on spectral flux information (Oliveira et al., 2010). Also, in the context of body movement, spectral flux has been associated with the type and temporal acuity of synchronization between the body and music at the beat rate (Burger et al., 2018) to a greater extent than other acoustic characterizations of musical rhythmic structure. As such, we found that spectral flux synchronized brain activity better than the amplitude envelope.”

p. 25, lines 497-498: "…we found…envelope." It is not clear if spectral flux causes stronger entrainment given the data. Rephrase to: "…we found that spectral flux was tracked better than the amplitude envelope."

Thank you. We changed the word “entrainment” to “synchronization” throughout the manuscript. For more consistency, we therefore rephrased the sentence to:

p. 27 l. 510-511: “As such, we found that spectral flux synchronized brain activity better than the amplitude envelope.”

p. 26, lines 520-521: "…and ease of beat tapping…stimulation tempi." The statistical analysis of the data in Figure 3 showed that ease of beat tapping did not significantly vary with stimulation tempo (lines 327-329), which contradicts this statement.

Yes, it is correct that we did not see any significant differences of the behavioral ratings across tempo conditions. We removed the sentence from the manuscript.

p. 28, lines 532-535: "One is that our study…entrainment." I don't understand why this would matter, because the analysis was done after selecting "musical" and "non-musical" subjects. Perhaps cut this?

We removed the “musicians” vs. “non-musicians” part from the manuscript due to the suggestions of Reviewer 2. Therefore, we cut this sentence.

p. 29, line 594: "correlate" should be "correlated".

Thank you. Done.

p. 32, line 647-648: "Each segment…0.25 Hz." The details for the range of tempi are in Supplementary Table 1, but it would be useful here to state the maximum amount of tempo shift relative to the original, both for increasing and decreasing the tempo. Additionally, I mentioned in the public review that the original tempos seem to span the range with the best neural tracking. You should also state the original range of tempos here.

We added the original music tempo range and maximum amount of tempo change as histograms to Figure 1 —figure supplement 2. We also added statements of the original music tempo in the Results section (p. 13 l. 265-273) and mention that it coincides with the tempo range of highest neural synchronization in the discussion (p. 23-24 l. 427-436).

p. 34, lines 703-704: "This signal…AFz position." Why did you choose to reference this way? Is there a related citation?

We chose to reference this way because the FCz and AFz are the standard positions for grounding and referencing the electrodes when using the actiCAP 64Ch Standard-2 system and layout from Brain Products. As these central positions are not ideal when conducting auditory experiments, we re-referenced our data to the average reference. This procedure has been used previously such as in Falk et al., 2017 and Cabral-Calderin and Henry, 2022.

p. 37, line 773: "To statistically asses" should be "assess".

Done. (Oops.)

p. 40, section EEG – Reliable Components Analysis, Temporal Response Function: In the public review, I suggested normalizing the measures of correlation, coherence, and prediction accuracy by null distributions. When you do so, make sure the shuffling of trials is done using trials with the same tempi. Otherwise, there will be a mismatch in the spectral content of the music due to differences in tempi, which defeats the point of generating the null distributions.

When calculating the z-score of the data, we took care that we calculated the surrogate distribution per tempo and participant. Details of the implementation can be found in the Materials and methods section:

p. 39 l. 823-831: “In order to control for any frequency-specific differences in the overall power of the neural data that could have led to artificially inflated observed neural synchronization at lower frequencies, the SRCorr and SRCoh values were z-scored based on a surrogate distribution (Zuk et al., 2021). Each surrogate distribution was generated by shifting the neural time course by a random amount relative to the musical feature time courses, keeping the time courses of the neural data and musical features intact. For each of 50 iterations, a surrogate distribution was created for each stimulation subgroup and tempo condition. The z-scoring was calculated by subtracting the mean and dividing by the standard deviation of the surrogate distribution.”

p. 40, line 855: "…a system identification technique…" This isn't quite true, change to "…a modeling technique…".

Thank you, we changed it.

p. 41, lines 873-884, also Figure 3C: From my understanding, TRFs were fit to each stimulus feature separately. I am ok with this for comparing prediction accuracies because the models appear to have the same dimensionality (T x 1, where T is the number of lags). However, this is potentially an issue for interpreting TRF weights because the stimulus features in the separate models are correlated, so peaks and troughs found in the TRF for one model might be the effect of one of the other features. One possibility is to look at the TRF weights of the full model instead, but ridge unfortunately optimizes the model by making the weights correlated (an effect that produces smooth TRFs), so I don't think it alleviates the issue either. The best approach would be to iteratively partial out the contribution of each stimulus feature. Start with the envelope model, compute the EEG prediction with that model, and then subtract the prediction from the EEG data. Then fit the derivative model and then the beats model, removing each after they are fit. This way, the TRF weights found for the spectral flux model reflect the temporal response after removing the effects of the other stimulus features.

Our mutual-information analysis showed that the musical features are indeed correlated, and in particular that spectral flux significantly shares mutual information with all other music features (Figure 1). Thus, the TRF weights for each feature will be necessarily nonindependent (though it’s not clear to us that they might be “an effect of one of the other features” alone). Therefore, we took you up on your analysis suggestion. However, we decided to not do a step-wise regression analysis, as this would lead to multiple orthogonalizations and there is no obvious reason in which order the TRFs should be computed. Instead, we calculated a multivariate TRF based on the amplitude envelope, first derivative of the envelope, and beat onsets (everything but spectral flux). Then, as you suggested, we subtracted the resulting predictions from the EEG data. The residual data were used to compute the TRF weights in response to spectral flux. The consequent TRF weights look qualitatively similar to the originals (compare to Figure 3F) and we added them to the analysis (Figure 3 —figure supplement 2):

We also plotted the TRF amplitude in the previously computed significant time lag window (102-211ms):

**Author response image 3. sa2fig3:** 

p. 42-43, lines 903-933: In the public review, I mentioned that the TRF-SVM modeling needed to be clearer. Specifically:– How are the TRFs fit for each group? If you are using a leave-one-trial-out procedure like before, the resulting TRFs for each trial are the models fit to all trials except the testing trial. As a result, they are going to be highly similar to each other, which could explain why the SVM performs so well. Here, I recommend instead fitting the TRF to each trial separately (no cross-validation) using the same ridge parameter that you found for the model trained earlier (in the section EEG – Temporal Response Function). That way the TRFs will represent the stimulus-response mappings for each individual trial.– Clarify which training step was referred to for calculating the surrogate data (line 920). I recommend shuffling prior to the TRF modeling step, although if you fit the TRF to each trial individually then the result of shuffling before vs after will be the same.

Thank you for making this important comment. We used a leave-one-trial-out procedure as before, and therefore agree that this way of analyzing the data is not the most suitable. In response to your comments we, as you suggest, either calculate the TRFs based on individual trials or calculate the TRF surrogate dataset by shuffling the labels prior to the TRF analysis (instead of implementing it at the training step of the SVM classifier). However, in neither of those cases the SVM accuracies of the actual data were significantly better than the SVM accuracies to a surrogate dataset. Therefore, we removed this part from the manuscript.

– Lines 913-916: There are 6 trials at tapped rate, and 6 trials at 2x tapped rate, which should result in n=12 not n=13.

By “n” we did not mean the number of trials, but rather how many participants and tempo conditions fulfilled this requirement. However, as we removed all SVM-related things from the manuscript, we also removed this part from the manuscript.

Figure 5 – supplement 1: The distributions on the left don't look like they come from the points on the right in each plot. Can you check if there was a mistake with the points plotted or the y-axes?

Thank you for this important comment. The left plot depicts the mean TRF correlations per participant whereas the right plot displayed the maximum TRF correlations per participant. Based on the comments of Reviewer 2, we removed the left plots (contrasting musicians vs. non-musicians) in each panel and only plotted the general sophistication index (from the Gold-MSI) against the TRF correlations. To make it more consistent with the Figures of the main manuscript we changed the maximum TRF correlations into the mean TRF correlations per participant (this way the right plot corresponds to the previous “musician vs. non-musician plot”, Figure 5 —figure supplement 2).

References

Burger, B., Ahokas, R., Keipi, A. and Toiviainen, P. (Year) Relationships between spectral flux, perceived rhythmic strength, and the propensity to move. City.

Cabral-Calderin, Y. and Henry, M.J. (2022) Reliability of Neural Entrainment in the Human Auditory System. J Neurosci, **42**, 894-908.

Crosse, M.J., Di Liberto, G.M., Bednar, A. and Lalor, E.C. (2016) The Multivariate Temporal Response Function (mTRF) Toolbox: A MATLAB Toolbox for Relating Neural Signals to Continuous Stimuli. Frontiers in Human Neuroscience, **10**.

Crosse, M.J., Zuk, N.J., Di Liberto, G.M., Nidiffer, A.R., Molholm, S. and Lalor, E.C. (2021) Linear Modeling of Neurophysiological Responses to Speech and Other Continuous Stimuli: Methodological Considerations for Applied Research. Frontiers in Neuroscience, **15**.

Di Liberto, G.M., Pelofi, C., Bianco, R., Patel, P., Mehta, A.D., Herrero, J.L., de Cheveigné, A., Shamma, S. and Mesgarani, N. (2020) Cortical encoding of melodic expectations in human temporal cortex. *eLife*, **9**, e51784.

Di Liberto, Giovanni M., O’Sullivan, James A. and Lalor, Edmund C. (2015) Low-Frequency Cortical Entrainment to Speech Reflects Phoneme-Level Processing. Current Biology, **25**, 2457-2465.

Ding, N., Chatterjee, M. and Simon, J.Z. (2014) Robust cortical entrainment to the speech envelope relies on the spectro-temporal fine structure. NeuroImage, **88**, 41-46.

Edwards, E. and Chang, E.F. (2013) Syllabic (∼2–5 Hz) and fluctuation (∼1–10 Hz) ranges in speech and auditory processing. Hearing Research, **305**, 113-134.

Ellis, R.J. and Jones, M.R. (2009) The role of accent salience and joint accent structure in meter perception. Journal of experimental psychology. Human perception and performance, **35 1**, 264-280.

Falk, S., Lanzilotti, C. and Schön, D. (2017) Tuning Neural Phase Entrainment to Speech. Journal of Cognitive Neuroscience, **29**, 1378-1389.

Jones, M.R. (1987) Dynamic pattern structure in music: Recent theory and research. Perception and Psychophysics, **41**, 621-634.

Jones, M.R. and Pfordresher, P.Q. (1997) Tracking musical patterns using joint accent structure. Canadian Psychological Association, Canada, pp. 271-291.

Kaneshiro, B., Nguyen, D.T., Norcia, A.M., Dmochowski, J.P. and Berger, J. (2020) Natural music evokes correlated EEG responses reflecting temporal structure and beat. NeuroImage, **214**, 116559.

Madsen, J., Margulis, E.H., Simchy-Gross, R. and Parra, L.C. (2019) Music synchronizes brainwaves across listeners with strong effects of repetition, familiarity and training. Scientific Reports, **9**, 3576.

Rajendran, V., Harper, N. and Schnupp, J. (2020) Auditory cortical representation of music favours the perceived beat. Royal Society Open Science, **7**, 191194.

Teng, X., Meng, Q. and Poeppel, D. (2021) Modulation Spectra Capture EEG Responses to Speech Signals and Drive Distinct Temporal Response Functions. eneuro, **8**, ENEURO.0399-0320.2020.

Vanden Bosch der Nederlanden, C.M., Joanisse, M.F., Grahn, J.A., Snijders, T.M. and Schoffelen, J.-M. (2022) Familiarity modulates neural tracking of sung and spoken utterances. NeuroImage, **252**, 119049.

Zuk, N.J., Murphy, J.W., Reilly, R.B. and Lalor, E.C. (2021) Envelope reconstruction of speech and music highlights stronger tracking of speech at low frequencies. PLOS Computational Biology, **17**, e1009358.